# A *Wolbachia* factor for male killing in lepidopteran insects

Susumu Katsuma [1,5] ✉, Kanako Hirota [1,5], Noriko Matsuda-Imai [1,5], Takahiro Fukui [1], Tomohiro Muro [1], Kohei Nishino[2], Hidetaka Kosako [2], Keisuke Shoji [3], Hideki Takanashi[1], Takeshi Fujii [4], Shin-ichi Arimura[1] & Takashi Kiuchi [1]

Bacterial symbionts, such as *Wolbachia* species, can manipulate the sexual development and reproduction of their insect hosts. For example, *Wolbachia* infection induces male-specific death in the Asian corn borer *Ostrinia furnacalis* by targeting the host factor Masculinizer (Masc), an essential protein for masculinization and dosage compensation in lepidopteran insects. Here we identify a *Wolbachia* protein, designated Oscar, which interacts with Masc via its ankyrin repeats. Embryonic expression of Oscar inhibits Masc-induced masculinization and leads to male killing in two lepidopteran insects, *O. furnacalis* and the silkworm *Bombyx mori*. Our study identifies a mechanism by which *Wolbachia* induce male killing of host progeny.

*Wolbachia* are the most widespread intracellular bacteria that alter host insect reproduction. *Wolbachia* enhance the production of infected female progeny via parthenogenesis, feminization, cytoplasmic incompatibility, or male killing. Each of these manipulations is considered to be adaptive for *Wolbachia* and facilitates their propagation[1,2]. *Wolbachia* factors resulting in cytoplasmic incompatibility in *Drosophila melanogaster* have recently been identified[3,4], but the genes underlying the other three manipulations remain largely unknown. Recent comparative genomics approach identified a candidate gene for male killing, which is conserved in several male-killing *Wolbachia* strains[5]. The candidate gene, *WO-mediated killing* (*wmk*), encodes a putative transcription factor with two helix-turn-helix DNA binding domains and is found within the eukaryotic association module of *Wolbachia* prophage WO in several *Wolbachia* strains as are the cytoplasmic incompatibility genes *cifA* and *cifB*[3,4]. Transgenic expression of codon-optimized *wmk* in *D. melanogaster* induces several cytological defects resulting in embryonic death and causes a significant female bias (the average male:female sex ratio is 0.65:1)[5]. Although the mode of action of *wmk* is unknown, Perlmutter et al.[5]. hypothesize that *wmk* is a strong candidate for the male-killing factor of several *Wolbachia* strains. *Wolbachia*-induced male killing is also

frequently observed in lepidopteran insects (i.e., butterflies and moths). In *Ostrinia scapulalis* and its congener *O. furnacalis*, a unique form of *Wolbachia*-induced male killing has been reported[6,7]. Unlike male killing observed in other butterflies and moths[8,9], *Wolbachia* depletion from infected strains with antibiotic treatment results in the production of all-male offspring[6,7]. This indicates that these *Wolbachia* possess a factor that induces feminization or inhibits masculinization in *Ostrinia* moths. Moreover, the *Ostrinia* feminizing factor itself may have been disrupted or inhibited during a prolonged period of infection by these male-killing *Wolbachia*.

Since failure of dosage compensation during development likely leads to sex-specific lethality in insects, the factors involved in this process are considered ideal targets for symbiont-induced sexual manipulation. The bacterial symbiont *Spiroplasma poulsonii* induces male killing in *D. melanogaster* by targeting the dosage compensation complex[10]. Similarly, we have shown that a male-killing *Wolbachia* targets a protein called Masculinizer (Masc), which is required for dosage compensation in lepidopteran insects[6]. Masc was discovered in the silkworm, *Bombyx mori*, as a factor required for both masculinization and dosage compensation[11]. Knockdown of *B. mori Masc* (*BmMasc*) inhibits masculinization and induces male-specific death

[1]Department of Agricultural and Environmental Biology, Graduate School of Agricultural and Life Sciences, The University of Tokyo, 1-1-1 Yayoi, Bunkyo-ku, Tokyo 113-8657, Japan. [2]Division of Cell Signaling, Fujii Memorial Institute of Medical Sciences, Institute of Advanced Medical Sciences, Tokushima University, Tokushima 770-8503, Japan. [3]Institute for Quantitative Biosciences, The University of Tokyo, 1-1-1 Yayoi, Bunkyo-ku, Tokyo 113-0032, Japan. [4]Faculty of Agriculture, Setsunan University, 45-1 Nagaotoge-cho, Hirakata, Osaka 573-0101, Japan. [5]These authors contributed equally: Susumu Katsuma, Kanako Hirota, Noriko Matsuda-Imai. ✉e-mail: skatsuma@g.ecc.u-tokyo.ac.jp

due to failure of dosage compensation in male embryos[11]. This is likely a phenocopy of male-specific embryonic death observed in *Wolbachia*-infected *Ostrinia* moths[6,7]. We previously observed inhibition of masculinization and failure of dosage compensation in *O. furnacalis* embryos infected with the male-killing *Wolbachia* *w*Fur[6]. Moreover, injection of artificially synthesized *O. furnacalis Masc* (*OfMasc*) cRNA rescues male killing in *w*Fur-infected *O. furnacalis* embryos[6]. These findings show that *w*Fur induces male killing in *O. furnacalis* embryos by targeting OfMasc, but it remains unknown how *w*Fur inhibits the OfMasc-induced signaling cascades. In this study, we identify and characterize a *Wolbachia* factor for male killing in lepidopteran insects.

## Results

### *Wolbachia* infection inhibits OfMasc accumulation in *O. furnacalis* cells

Since *Wolbachia* propagate in the cytoplasm, we hypothesized that *w*Fur itself or *w*Fur-derived factor(s) directly interacts with OfMasc in the cytoplasm and inhibits OfMasc function in both the cytoplasm and nucleus. To test this hypothesis, we established a cell-based system for the biochemical identification/analysis of key proteins involved in male killing. We first generated cell lines from *O. furnacalis* embryos infected with *w*Fur (Supplementary Fig. 1a). We observed the presence of *w*Fur in the cytoplasm (Fig. 1a, b, Supplementary Fig. 1b) and confirmed that tetracycline treatment completely eliminated *w*Fur from

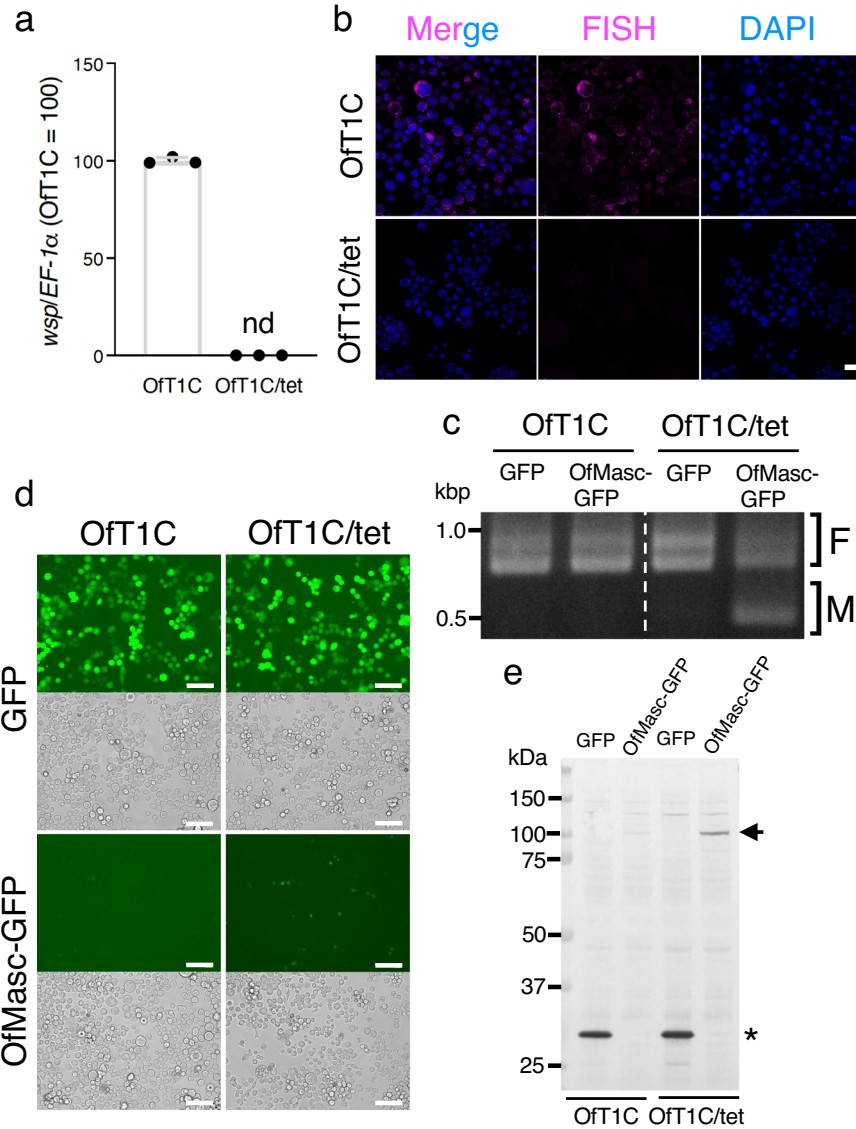

**Fig. 1 | Establishment of cell lines from *w*Fur-infected *O. furnacalis* embryos.** **a** *Wolbachia* density in OfT1C and OfT1C/tet cells. *Wolbachia* density was estimated by qPCR of *wsp* and normalized by *EF-1α*. Data shown are means ± SD of triplicate measurements. Similar results were obtained in two independent experiments. nd, not detected. **b** FISH analysis of *Wolbachia* in OfT1C and OfT1C/tet cells. *Wolbachia* were detected by a 16 S rRNA FISH probe. The cells were also stained with DAPI for DNA. Bar, 20 μm. Similar results were obtained in two independent experiments. **c** Splicing patterns of *Ofdsx*. OfT1C and OfT1C/tet cells were transfected with *GFP* or *OfMasc-GFP* cDNA, and *Ofdsx* splicing was investigated. The F and M indicate female- and male-type splicing of *Ofdsx*, respectively. Similar results were obtained in two independent experiments. **d** Fluorescence microscopy of OfT1C and OfT1C/tet cells transfected with *GFP* or *OfMasc-GFP* cDNA. Bar, 100 μm. Similar results were obtained in two independent experiments. **e** Immunoblots of GFP and OfMasc-GFP in OfT1C and OfT1C/tet cells. Cells were transfected with *GFP*- or *OfMasc-GFP*-expressing plasmids, harvested at 3 days after transfection, and subjected to immunoblotting. Arrow and asterisk indicate the positions of OfMasc-GFP and GFP, respectively. Similar results were obtained in three independent experiments and summarized in Supplementary Fig. 2a. Source data are provided as a Source Data file.

these cells (Fig. 1a, b). In the cell lines OfT1C and *w*Fur-eliminated OfT1C (OfT1C/tet) (Supplementary Fig. 1a), we observed a female-specific default splicing pattern of *O. furnacalis doublesex* (*Ofdsx*)[6,12], a gene which commonly acts at the downstream end of the sex differentiation cascade in insects[13] (Fig. 1c, Supplementary Fig. 1c). Transfection of OfT1C/tet cells with *OfMasc* or *GFP*-fused *OfMasc* (*OfMasc-GFP*) cDNA induced the expression of male-type *Ofdsx* variant (*Ofdsx^M*) (Fig. 1c, Supplementary Fig. 1c). However, as observed in male larvae hatched from *OfMasc* cRNA-injected *w*Fur-infected embryos[12], OfMasc expression did not affect the splicing pattern of *Ofdsx* in OfT1C cells (Supplementary Fig. 1c, Fig. 1c). Moreover, GFP fluorescence was only faintly observed in *OfMasc-GFP*-transfected OfT1C cells (Fig. 1d). Similar results were obtained in other cell lines OfT1B and OfT1A-Ki (Supplementary Fig. 1d–f). OfT1A-Ka, whose origin is the same as OfT1A-Ki but shows significantly lower intracellular *Wolbachia* density than OfT1A-Ki (Supplementary Fig. 1g), exhibited GFP fluorescence when transfected with *OfMasc-GFP* (Supplementary Fig. 1g). Immunostaining of OfT1C cells also revealed that OfMasc accumulated at high levels in cells with low *Wolbachia* density (Supplementary Fig. 1h). These results suggest that OfMasc accumulation in *O. furnacalis* cells is negatively correlated with *Wolbachia* density in individual cells. Immunoblot experiments showed a severe reduction of OfMasc accumulation in OfT1C cells compared with that in OfT1C/tet cells (Fig. 1e, Supplementary Fig. 2a). This decrease was inhibited by the addition of PS-341 (Bortezomib), a proteasome inhibitor, into the culture medium without affecting intracellular *Wolbachia* density (Supplementary Fig. 2b, c). This suggested that reduced OfMasc accumulation in *w*Fur-infected OfT1C cells is involved in protein degradation via the proteasome system. In addition, these results indicated that *w*Fur infection inhibits OfMasc accumulation, presumably resulting in the inhibition of OfMasc-induced masculinization in *O. furnacalis* cultured cells.

**Identification of a *Wolbachia* protein that interacts with OfMasc**

Our experiments showed that the newly established *O. furnacalis* cultured cells likely mimic the phenotypes of *w*Fur-infected *O. furnacalis* embryos. We next generated four *OfMasc-GFP* derivatives (Supplementary Fig. 3a) in order to identify the region of OfMasc that interacts with *w*Fur-derived factors in the cytoplasm. We found that OfMasc lacking 87 N-terminal amino acid residues (Del1-OfMasc-GFP) efficiently accumulated in OfT1C cells (Fig. 2a, Supplementary Fig. 3b, c). These results indicated that the N-terminal region is important for strong interaction with *w*Fur-derived factor(s) in the cytoplasm. We next attempted to identify OfMasc-interacting *w*Fur proteins by comparing immunoprecipitates with anti-GFP nanobody from *GFP*-, *OfMasc-GFP*-, and *Del1-OfMasc-GFP*-transfected OfT1C cells. Liquid chromatography-tandem mass spectrometry (LC-MS/MS) analyses identified 1,045 host- and *w*Fur-derived proteins (Fig. 2b, Supplementary Data 2). Among these, we discovered a single *w*Fur protein that specifically and reproducibly binds OfMasc via the N-terminal region (Fig. 2b, c). This interaction was verified by co-immunoprecipitation experiments of OfT1C/tet cells co-transfected with *3×FLAG-tagged* this gene and *GFP*, *OfMasc-GFP* or *Del1-OfMasc-GFP* (Supplementary Fig. 3d). We named this OfMasc-interacting *w*Fur protein Oscar (**Os**ugoroshi protein containing **C**ifB C-terminus-like domain and many **A**nkyrin **R**epeats; Osugoroshi translates to male killing in Japanese).

The N-terminal region (a.a. 2–88) of OfMasc contains two CCCH-type zinc fingers, ZF1 and ZF2 (Supplementary Fig. 3a). To examine the importance of two zinc fingers for Oscar–OfMasc interaction, we generated two constructs expressing OfMasc-C38A-GFP (ZF1 mutant) or OfMasc-C70A-GFP (ZF2 mutant). GFP fluorescence and immunoblot experiments revealed that ZF1 plays a more dominant role than ZF2 (Supplementary Fig. 3e, f). LC-MS/MS-based targeted quantification

also supported the importance of ZF1 for Oscar–OfMasc interaction (Supplementary Fig. 3g).

Oscar is an 1830 amino acid residue-long protein with a predicted molecular mass of 201.1 kDa (Supplementary Fig. 4). This size is consistent with an endogenous Oscar protein detected in OfT1C cells as well as in vitro translated Oscar protein (Supplementary Fig. 5a, b). Oscar possesses a CifB-like domain at the C-terminus and 40 copies of the ankyrin repeat motif (Fig. 2d, Supplementary Fig. 4). CifB is a cytoplasmic incompatibility factor identified in other cytoplasmic incompatibility-inducing *Wolbachia* and the CifB C-terminal region contains a ubiquitin-like-specific protease 1 domain (Ulp1)[3,4,14], suggesting that Oscar possibly possesses Ulp1 activity. The ankyrin repeat is a motif that contains two alpha helices, and its tandem repeats mediate protein–protein interactions[15,16]. Ankyrin repeats are most commonly found in eukaryote proteins, but are also present in some bacterial effectors such as the *S. poulsonii* male-killing protein Spaid[10,15]. AlphaFold2 prediction software suggested that the ankyrin repeat region (a.a. 1–1370) and the linker/CifB C-terminus-like region (a.a. 1371–1830) of Oscar both fold into compact structures (Fig. 2e, Supplementary Fig. 5c). In particular, the ankyrin repeat region containing 40 ankyrin repeats was predicted to form an elongated, superhelical structure (Fig. 2e).

To predict the complex structure of Oscar and OfMasc, we first investigated the structures of Masc proteins. Previous studies revealed that the C-terminal 288 amino acid residues (a.a. 301–588) of BmMasc is sufficient for its masculinizing activity and required for dosage compensation[17,18]. In this masculinizing region, two cysteine residues at 301 and 304 are crucial for its activity[17]. AlphaFold2 prediction suggested that the BmMasc sequence is generally unstructured except for two ZFs and the masculinizing region. The masculinizing region followed by two cysteine residues is predicted to form a long α-helix (a.a. 305–349, Supplementary Fig. 5d). OfMasc was predicted to form a similar structure to that of BmMasc with the α-helix of the masculinizing region (a.a. 257–301) (Fig. 2e). The predicted complex formed between the ankyrin repeat region of Oscar (a.a. 1–1,370) and OfMasc suggested that the cavity formed by the superhelical structure of the ankyrin repeats of Oscar captures OfMasc. In this complex, the α-helix of the OfMasc masculinizing domain was drastically changed (Fig. 2e), suggesting that Oscar binding to OfMasc may inhibit its function by disrupting the essential structure of OfMasc.

We identified an *Oscar* homolog only in the draft genome of *w*Bol-1b, a male-killing *Wolbachia* strain of the butterfly *Hypolimnas bolina*[19]. However, only a single ankyrin repeat was identified in this homolog, presumably because the genome assembly of *w*Bol-1b is still incomplete (Supplementary Fig. 6a). The *w*Fur genome around the *Oscar* locus showed a high-level synteny with that of *w*Pip (Supplementary Fig. 6b). This *w*Pip region is not annotated as the prophage WO region that frequently contains *Wolbachia* genes for sexual manipulation[3,5]. The *w*Pip genome possesses a gene that exhibits high homology to the C-terminal region of Oscar CifB C-terminus-like domain (Supplementary Fig. 6b). These results suggested that the ankyrin repeat region and most of the CifB C-terminus-like domain has been deleted from the genome of an ancestral *Wolbachia* of *w*Pip. In contrast, lepidopteran *Wolbachia* belonging to Supergroup B (e.g., *w*Fur and *w*Bol-1b) have maintained *Oscar* homologs in their genomes and is predicted to use them for male killing. We experimentally detected the full-length amplicons of *Oscar* from DNAs of *w*Fur-infected *O. furnacalis* and male-killing *Wolbachia*-infected *O. scapulalis*, but not from those of uninfected *O. furnacalis* and *O. scapulalis*, all of which are maintained in our laboratory. In addition, we detected *Oscar* fragments from DNAs of field-collected *Ostrinia* species that were infected with male-killing *Wolbachia* (Supplementary Fig. 6c). These findings suggest that *Oscar* is commonly present in the genomes of male-killing *Wolbachia* infecting *Ostrinia* species.

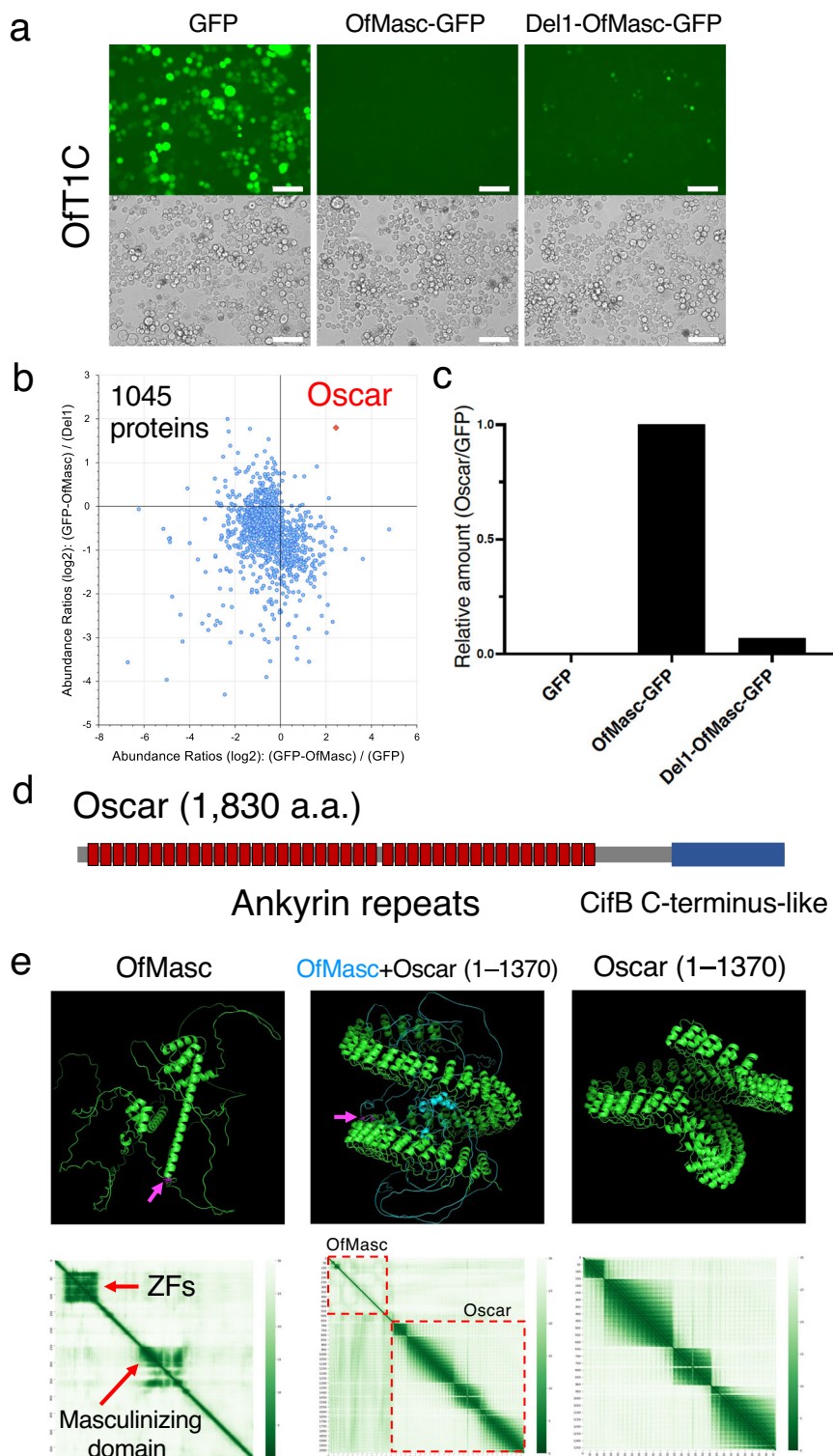

**Fig. 2 | Identification of Oscar as the OfMasc-interacting protein. a** Fluorescence microscopy of OfT1C cells transfected with *GFP*, *OfMasc-GFP*, or *Del1-OfMasc-GFP* cDNAs. Bar, 100 μm. Similar results were obtained in three independent experiments. **b** LC-MS/MS analysis of immunoprecipitates with anti-GFP nanobody from *GFP-*, *OfMasc-GFP-*, or *Del1-OfMasc-GFP*-transfected OfT1C cells. Dots indicate host and *w*Fur proteins identified by label-free precursor ion quantification (1045 proteins). Oscar is indicated by the red dot. Similar results were obtained in two independent experiments. **c** Parallel reaction monitoring (PRM) quantification of relative amount of Oscar to GFP in *GFP-*, *OfMasc-GFP-*, or *Del1-OfMasc-GFP-* transfected OfT1C cells. **d** Structure of Oscar. Ankyrin repeats and CifB C-terminus-like domain are shown. **e** Structure prediction of the OfMasc–Oscar complex. Structures of OfMasc, OfMasc–Oscar (residues 1–1370) complex, and Oscar (residues 1–1370) were predicted by AlphaFold2. Two cysteine residues at 253 and 256 of OfMasc are shown in magenta (also indicated by magenta arrows). The heatmaps of predicted aligned errors (PAEs) are also shown below each model. PAEs indicate the expected positional error at residue *x* if the predicted and actual structures are aligned on residue *y*. AlphaFold2 has high confidence when visualized in green. Source data are provided as a Source Data file.

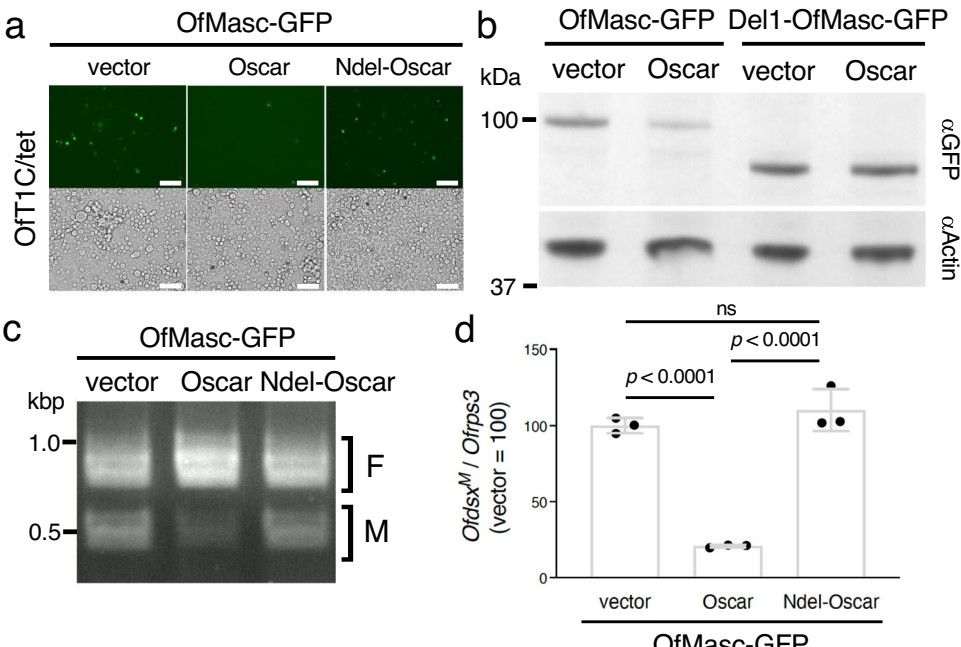

**Fig. 3 | Oscar inhibits OfMasc accumulation and OfMasc-induced masculinization in cultured cells. a** Fluorescence microscopy of OfT1C cells co-transfected with *OfMasc-GFP* and empty vector or *Oscar* cDNAs. Bar, 100 μm. Similar results were obtained in two independent experiments. **b** Accumulation of OfMasc-GFP or Del1-OfMasc-GFP in Oscar-expressed OfT1C/tet cells. Actin was used as a normalization control. Similar results were obtained in two independent experiments. **c** Splicing patterns of *Ofdsx*. OfT1C/tet cells were co-transfected with *OfMasc-GFP* and empty vector or *Oscar* cDNAs, and *Ofdsx* splicing was investigated. Similar results were obtained in two independent experiments. **d** Quantification of the male-type splice variants of *Ofdsx* ($Ofdsx^M$). $Ofdsx^M$ levels were estimated by RT-qPCR. Data shown are means ± SD of triplicate measurements. Adjusted *p* values following one-way ANOVA with Tukey's multiple comparisons tests are shown. Similar results were obtained in two independent experiments. ns, $p > 0.05$. Source data are provided as a Source Data file.

## Oscar induces male killing in moths

We next investigated the function of Oscar in cultured cells. Transfection of codon-optimized *Oscar* cDNA inhibited OfMasc-GFP fluorescence in OfT1C/tet cells (Fig. 3a). This Oscar-induced inhibition is likely a phenocopy of that found in *w*Fur-infected *O. furnacalis* cultured cells (Fig. 1d, Supplementary Fig. 1d, f). The inhibitory function was also observed in *Oscar*-transfected *Spodoptera frugiperda*-derived Sf-9 and *B. mori*-derived BmN-4 cells (Supplementary Fig. 7a, b). In addition, immunoblot experiments showed reduced accumulation of OfMasc-GFP in Oscar-expressing OfT1C/tet cells (Fig. 3b). This reduction was not observed in *Del1-OfMasc-GFP*-transfected OfT1C/tet cells (Fig. 3b), confirming the importance of the OfMasc N-terminus for interaction of Oscar and OfMasc. Treatment of Oscar-expressing OfT1C/tet cells with PS-341 inhibited the decrease in OfMasc-GFP accumulation (Supplementary Fig. 7c). This suggested that OfMasc is degraded even in the absence of *Wolbachia* infection, via the proteasome pathway after forming a complex with Oscar. Moreover, we found that Oscar expression markedly reduces the amount of the male-type *dsx* splice variant in OfMasc-GFP-expressing OfT1C/tet, Sf-9, and BmN-4 cells (Fig. 3c, d, Supplementary Fig. 7d, e). These results suggested that Oscar alone inhibits OfMasc-induced masculinization in *O. furnacalis* and other lepidopteran cultured cells by decreasing OfMasc accumulation. The N-terminal deleted Oscar (Ndel-Oscar, Δ2–951), in which 29 copies of ankyrin repeats are missing, affected neither OfMasc-GFP accumulation nor OfMasc-induced masculinization compared to control cells (Fig. 3a, c, Supplementary Fig. 7a, b, d, e), indicating that more than 11 copies of ankyrin repeats are required for Oscar function.

We generated four additional deletion constructs, each of which expresses a mutant Oscar protein (Supplementary Fig. 8a), and assessed their activities in Sf-9 cells. We first examined whether Oscar derivatives were successfully expressed. We observed similar expression levels of DelA and DelB compared to that of wild-type Oscar. We

could not detect DelC and DelD expression with anti-Oscar antibody, because this antibody was raised against a synthetic peptide corresponding to the C-terminus of Oscar (a.a. 1,512–1,525). We generated additional *3×FLAG-tagged Oscar* constructs (*FLAG-DelC* and *FLAG-DelD*) and detected their expressions using anti-FLAG antibody (Supplementary Fig. 8c). GFP fluorescence and immunoblot experiments revealed that the number of ankyrin repeats is positively correlated with Oscar's inhibitory activity against OfMasc accumulation (Supplementary Fig. 7a, 8b, 8c, 8d) and OfMasc-induced masculinization (Supplementary Fig. 7d, 8e, 8f). The degree of Oscar's activity decreased in the following order: (1) wild-type (40 ankyrin repeats); (2) DelA (33 ankyrin repeats); (3) DelB (23 ankyrin repeats); and (4) Ndel-Oscar (11 ankyrin repeats). We also found that CifB-like C-terminus domain is not essential for Oscar's function, whereas the loss of both CifB-like domain and linker region drastically decreased Oscar's activity (Supplementary Fig. 8b–f). Structural prediction revealed that the linker region forms a long alpha-helix structure (Supplementary Fig. 5c), suggesting that this structure is crucial for Oscar's function. DelA and DelB both inhibited OfMasc-induced masculinization, although OfMasc accumulation was not greatly reduced (Supplementary Fig. 8b, 8d, 8e). This is presumably because these ankyrin repeat mutants bind OfMasc and inhibit its activity, whereas they do not lead to the proteasome-dependent degradation of OfMasc efficiently.

Expression of lepidopteran Masc proteins in BmN-4 cells induces the splicing of the male-type variants of *B. mori dsx* ($Bmdsx^M$) as well as *B. mori IGF-II mRNA-binding protein* ($BmIMP^M$), the product of which is involved in the male-specific splicing of *Bmdsx* (Supplementary Fig. 7d, e, Supplementary Fig. 9a, b)[11,12,19,20]. We observed that transfection of *Oscar* cDNA reduces the masculinizing activity of Masc proteins from five lepidopteran insects in BmN-4 cells (Supplementary Fig. 7e, Supplementary Fig. 9c, d), suggesting that Oscar commonly inhibits Masc-dependent signaling cascades in lepidopteran insects. To verify Oscar

function at the organismal level, we used *B. mori* and *O. furnacalis* embryos. We injected *GFP* (control) or *Oscar* cRNA into uninfected embryos, examined the splicing pattern of *dsx* in embryos, and sexed the hatched larvae molecularly. Injection of *Oscar* cRNA in male *B. mori* embryos inhibited the splicing of *Bmdsx*<sup>M</sup> as well as *BmIMP*<sup>M</sup> expression (Fig. 4a–c). The hatched larvae injected with *Oscar* cRNA were genetically all female, whereas both male and female larvae were observed when injected with *GFP* cRNA (Fig. 4d). Additionally, most of the dead embryos by *Oscar* cRNA injection were male (Fig. 4e). These results clearly indicate that Oscar inhibits masculinization and induces male killing in *B. mori*. We next examined *Ofdsx* splicing in *O. furnacalis* egg masses, all embryos of which were injected with *GFP* or *Oscar* cRNA. *O. furnacalis* embryos injected with *GFP* cRNA expressed female-type *Ofdsx* variant (*Ofdsx*<sup>F</sup>) and *Ofdsx*<sup>M</sup> equally, whereas *Oscar* cRNA injection inhibited *Ofdsx*<sup>M</sup> splicing (Supplementary Fig. 10a). Injection of *Oscar* cRNA led to female-biased progeny production (the male:female sex ratio is 0.33:1), whereas this bias was not observed in *GFP* cRNA-injected control embryos (Supplementary Fig. 10b). Moreover, the adult moths emerged from *Oscar* cRNA-injected embryos were all female, whereas both male and female moths were observed when injected with *GFP* cRNA (Supplementary Fig. 10c). These results indicate that Oscar expression mimics *Wolbachia*-induced male-specific death in *O. furnacalis* during embryonic[6] and larval stages[21,22]. These findings show that Oscar is the male-killing factor of *w*Fur.

## Discussion

Our present study answers the question of how *Wolbachia* accomplish male killing in lepidopteran insects. The male-killing factor Oscar interacts with Masc and reduces Masc accumulation, which subsequently inhibits masculinization and disrupts dosage compensation in host cells, finally leading to male-specific lethality (Fig. 4f). To the best of our knowledge, this is the first discovery of a *Wolbachia* male-killing factor whose mode of action is elucidated. Structure prediction analyses suggest that 40 copies of ankyrin repeats form a superhelical structure that captures Masc within its cavity (Fig. 2e). Several ankyrin repeat-containing proteins are known to serve critical roles in the ubiquitin-proteasome pathway[23]. In our experiments, *w*Fur infection or Oscar expression reduces OfMasc accumulation via the proteasome pathway (Supplementary Fig. 2b, Supplementary Fig. 7c). Considering that Oscar possesses an Ulp1-like domain at its C-terminus, Oscar itself may be involved in the degradation of captured proteins. *S. poulsonii* male-killing protein Spaid also has 4 copies of ankyrin repeats and a deubiquitylation domain[10], suggesting that ankyrin repeat-mediated protein–protein interactions and proteasome-dependent degradation may be commonly used in sexual manipulation by symbiotic bacteria in host insects.

Previous studies suggested that *wmk* is a strong candidate for *Wolbachia*'s male-killing factor[5,24]. Our study identified Oscar as a novel male-killing factor at least in lepidopteran insects, which does not show any sequence homology to Wmk (Fig. 2d). Since *w*Fur possesses two *wmk* genes and some *wmk*-like genes (Supplementary Fig. 11a), we investigated their functions in lepidopteran cells. We selected three *wmk* homologs (designated as *wmk1*, *wmk2*, and *wmk3*), which are closely related to *w*Mel *wmk* (Supplementary Fig. 11a), codon-optimized them, and examined the effects of their expression on OfMasc-induced masculinization in Sf-9 cells. We found that *w*Fur Wmks do not inhibit OfMasc accumulation (Supplementary Fig. 11b) and OfMasc-induced *Sfdsx*<sup>M</sup> splicing in Sf-9 cells (Supplementary Fig. 11c). These results suggest that *wmk* genes may play roles other than male killing in lepidopteran insects.

We and other group independently reported that infection with male-killing *Wolbachia* results in *Masc* mRNA decrease in *O. furnacalis*[6] and *O. scapulalis*[7] embryos, respectively. Combined with the result that injection of *OfMasc* cRNA rescues male killing in *w*Fur-infected *O. furnacalis* embryos[6], we hypothesized that male killing is caused by *OfMasc*

reduction at the mRNA level. In this study, however, we show that OfMasc is regulated at the protein level in *w*Fur-infected cells: binding of Oscar to OfMasc impairs OfMasc functions and finally induces proteasome-dependent degradation of OfMasc. Although we do not understand the mechanism and importance of *OfMasc* mRNA decrease in *w*Fur-infected *O. furnacalis* embryos, this phenomenon is likely an indirect result by a long-term infection with a male-killing *Wolbachia*.

In conclusion, our study reveals that *Wolbachia* have evolved to hijack the lepidopteran insect-specific sex determination and dosage compensation systems by taking advantage of Oscar, a novel protein that contains a huge number of ankyrin repeats. In *Ostrinia* species, removal of *Wolbachia* from male-killing strains results in female killing instead of restoring the 1:1 sex ratio[6,7]. This indicates that the *Ostrinia* feminizing gene has been disrupted or inactivated after the acquisition of *Oscar*. Genome or epigenome comparison between *w*Fur-infected and uninfected *O. furnacalis* will provide evidence for *Wolbachia*-induced loss of a host sex-determining gene and identify the *Ostrinia* feminizing factor that inhibits OfMasc-mediated signaling in females of uninfected strains.

## Methods

### Insects and cell lines

Larval *O. furnacalis* and *O. scapulalis* were reared on an artificial diet (Insecta LFS, Nosan Corp., Japan) at 25 °C under a photoperiod of 16L and 8D[6,12]. *O. furnacalis* and *O. scapulalis* moths used in this study were collected at Nishi-Tokyo, Japan (35.7° N, 139.5° E) in early summer of 2020 and 2021. *Wolbachia*-infected *O. furnacalis* moths were collected at Matsudo, Japan (35.8° N, 139.9° E) in early summer of 2014 and at Nishi-Tokyo, Japan (35.7° N, 139.5° E) in early summer of 2021, and *Wolbachia*-infected *O. scapulalis* moths were collected at Matsudo, Japan (35.8° N, 139.9° E) in early summer of 2020. Larval *B. mori* (p50T strain) was reared on an artificial diet (SilkMate PS, Nosan Corp., Japan) or mulberry leaves at 25 °C under a photoperiod of 18L and 6D.

*B. mori* BmN-4 cells (provided by Chisa Yasunaga-Aoki, Kyushu University, and maintained in our laboratory) and *S. frugiperda* Sf-9 cells (provided by Ryoichi Sato, Tokyo University of Agriculture and Technology, and maintained in our laboratory) were cultured at 26 °C in IPL-41 and TC-100 medium (Applichem, Germany) supplemented with 10% fetal bovine serum, respectively. *O. furnacalis* cell lines, OfT1A, OfT1B, and OfT1C, were established from *w*Fur-infected *O. furnacalis* embryos (see below) and cultured at 26 °C in Express Five™ SFM (Gibco, USA) supplemented with 18 mM L-Glutamine and 10% fetal bovine serum (FBS) (Gibco). OfT1C/tet and OfT1B/tet cells were cultured in the same medium containing 3 μg/mL tetracycline.

### Establishment of *O. furnacalis* cell lines

An egg mass obtained from *w*Fur-infected *O. furnacalis* female was collected into a 1.5 mL microfuge tube at 2–3 days post-oviposition and washed twice with 1 mL of phosphate-buffered saline (PBS). The egg surface was sterilized with 1 mL of 3% formaldehyde solution in PBS for 10 min and washed with PBS for 10 min using a tube rotator. After removal of the PBS, the egg mass was crushed gently using an autoclaved pestle in 100 μL of Express Five™ SFM supplemented with 18 mM L-Glutamine and 10% FBS. The crushed egg mass was transferred into a 35 mm-diameter culture dish containing 2 mL of Express Five™ SFM supplemented with 18 mM L-Glutamine and 10% FBS, 2% Penicillin-Streptomycin (10,000 U/mL) (Gibco), and 0.2% Amphotericin B (250 μg/mL) (Gibco) and cultured at 26 °C. After some cell clusters attached to the bottom of dish without any contamination, half of the spent medium was replaced with fresh medium at weekly intervals. After the cells showed obvious signs of proliferation, the cells were collected by pipetting, seeded to a new dish or tissue culture flask, and maintained separately in Express Five™ SFM supplemented with 18 mM L-Glutamine and 10% FBS, 1% Penicillin-Streptomycin, and 0.1% Amphotericin B.

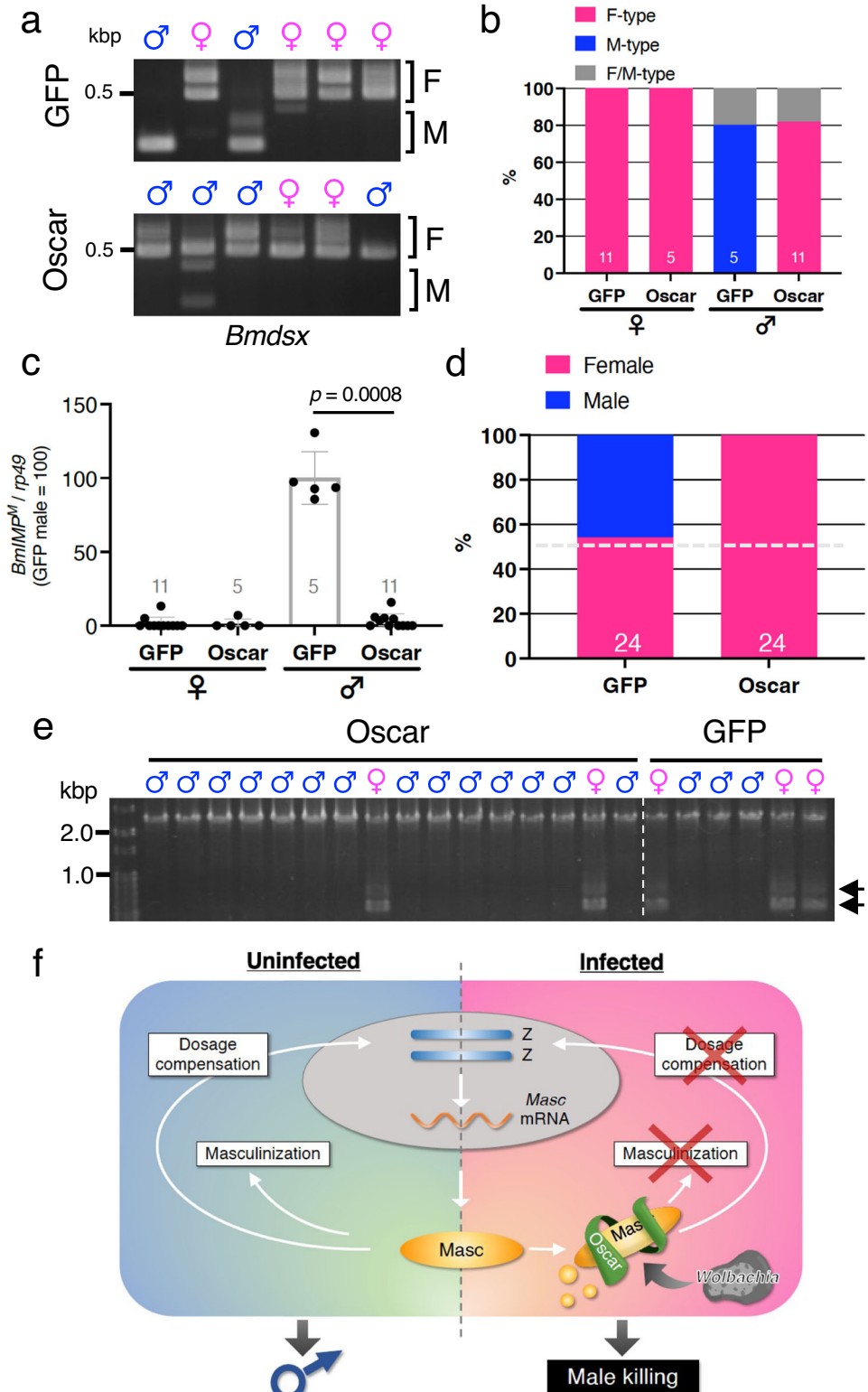

**Fig. 4 | Oscar inhibits Masc-induced masculinization and induces male killing.**
**a, b** *Bmdsx* splicing in *GFP* (*n* = 16) or *Oscar* (*n* = 16) cRNA-injected *B. mori* embryos at 48 h post-injection. Representative splicing patterns are shown in **a** and the data are summarized in **b**. The F and M indicate *Bmdsx$^F$ and Bmdsx$^M$*, respectively. The number indicates the sample size of each group. **c** *BmIMP$^M$* expression in *GFP* (*n* = 16) or *Oscar* (*n* = 16) cRNA-injected *B. mori* embryos at 48 h post-injection. Data shown are means ± SD. Adjusted *p* values following one-way ANOVA with Tukey's

multiple comparisons tests are shown. The number indicates the sample size of each group. **d** *GFP* (*n* = 24) or *Oscar* (*n* = 24) cRNA was injected into *B. mori* embryos immediately after oviposition. The hatched larvae were collected and molecularly sexed. The number indicates the sample size of each group. **e** Molecular sexing of all dead embryos by *Oscar* (*n* = 16) or *GFP* (*n* = 6) cRNA injection. The arrows indicate the positions of female-specific bands. **f** A proposed model for Oscar-mediated male killing in lepidopteran insects. Source data are provided as a Source Data file.

## Sex pheromone analysis by gas chromatography-mass spectrometry (GC-MS)

*Ostrinia* species were identified by visual appearance and sex pheromones of the virgin females. Sex pheromone components of *O. furnacalis* and *O. scapulalis* are mixtures of (E/Z)−12-tetradecenyl acetates (E/Z12-14:OAc) and (E/Z)−11-tetradecenyl acetates (E/Z11-14:OAc), respectively[25]. The hexane extract of a pheromone gland was individually analyzed using a gas chromatograph coupled to a mass spectrometer (QP2010 SE GC−MS, Shimadzu, Japan) equipped with a capillary column (DB-Wax, 0.25 mm i.d. × 30 m; Agilent Technologies, USA). The initial column oven temperature of 80 °C was held for 2 min, then raised at 8 °C/min to 240 °C and held for 2 min. The flow rate of the carrier gas (helium) was 1.0 mL/min. Mass spectra were recorded in the electron ionization mode at 70 eV. Retention time and diagnosis ions ($m/z$ 61 and $m/z$ 194 for [M- 60]$^+$; i.e., elimination of acetic acid) of peak were compared to authentic standard.

## Reverse transcription-PCR (RT-PCR)

Three days after transfection, total RNA was prepared from the transfected cells using TRI REAGENT® (Molecular Research Center Inc., USA). cDNA synthesis was carried out using 500 ng total RNA and avian myeloblastosis virus reverse transcriptase with an oligo-dT primer (TaKaRa, Japan). PCR Amplification of *Ofdsx* and *Bmdsx* cDNA was carried out by 3-step PCR (annealing temperature is 60 °C, 38 cycles) using KOD FX−neo DNA polymerase (TOYOBO, Japan) with published primers[6,11]. RT-PCR for *Sfdsx* was performed under same condition using the following primers:

Sfdsx_F2: 5′-ACCGCTGCCGCAAGCGATGC-3′
Sfdsx_R1: 5′-AATAATCGCCGATCGATATC-3′

The uncropped and unprocessed scans used for figures are provided in the Source Data file.

Amplified *Sfdsx* cDNAs were sequenced and deposited in GenBank. Quantitative RT-PCR (RT-qPCR) of *Sfdsx*$^M$ was performed by StepOne-Plus and StepOne Software v2.3 (Thermo Fisher Scientific, USA) using a KAPA SYBR FAST qPCR kit (Kapa Biosystems, USA) and the following primers:

Sfdsx_qF1M: 5′-GGAAAATAGACGAAGCCCAC-3′
Sfdsx_qR2: 5′-CGTACTCCGTGAAGCACATG-3′

The mRNA level was normalized to that of *Sfrp49* and expression values were calculated using the 2$^{-\Delta\Delta Ct}$ method. RT-qPCR of *Sfrp49* was performed using the following primers:

Sfrp49_qF1: 5′-CCCAACATTGGTTACGGATC-3′
Sfrp49_qR1: 5′-TTCTTTGAGGAGACTCCGTG-3′

RT-qPCR of *Ofdsx*$^M$ was performed using a KAPA SYBR FAST qPCR kit (Kapa Biosystems) and the following primers:

OfdsxM_F2: 5′-GAAGATTGATGAAGCCCACTG-3′
OfdsxM_R1: 5′-GCACTGTGTCTATCACACTG-3′

The mRNA level was normalized to that of *Ofrps3*[6] and expression values were calculated using the 2$^{-\Delta\Delta Ct}$ method.

RT-qPCR of *BmIMP*$^M$ and *rp49* was performed using the published primers[17].

## Estimation of *Wolbachia* density

Genomic DNA was prepared from *O. furnacalis* cultured cells using DNeasy Blood & Tissue Kit (QIAGEN, Germany). *Wolbachia* density was estimated by qPCR for *wsp*. qPCR was carried out using a KAPA SYBR FAST qPCR Kit (Kapa Biosystems Inc.) and the values were calculated using the 2$^{-\Delta\Delta Ct}$ method. The amplification values were normalized to those of *EF-1α*[6]. The qPCR primers for *wsp* were as follows:

Wsp_F: 5′-GAAACAAATGTTGCAGACAG-3′
Wsp_R2: 5′-AACTGTATCAGCTTTTGAAG-3′.

## Plasmid construction and transfection

Mutagenesis and deletion experiments were conducted using the KOD-Plus mutagenesis kit (TOYOBO, Japan), In-Fusion HD Cloning Kit (Clontech, USA), or by restriction enzyme digestion. The codon-optimized *Oscar*, *wmk1*, *wmk2*, and *wmk3* fragments were synthesized by Eurofins Genomics (Germany). The DNA fragments of *OfMasc-GFP*[26], *wmk1*, *wmk2*, *wmk3*, *Oscar*, and their derivatives were cloned into the pIZ/V5-His-g3 vector[26]. *Trilocha varians Masc* (*TvMasc*)[27], *S. frugiperda Masc* (*SfMasc*), and *Papilio machaon Masc* (*PmMasc*) cDNAs fused to *EGFP* fragment were also cloned into the pIZ/V5-His-g3 vector. The cDNA fragment of *SfMasc* was cloned from Sf-9 cells and determined by Sanger sequencing. *PmMasc* cDNA was codon-optimized and synthesized by Eurofins Genomics. The sequences of artificially synthesized cDNA were listed in Supplementary Data 1.

Cultured cells ($4 \times 10^5$ cells per 35-mm diameter dish) were transfected with 1 μg of plasmid DNAs using FuGENE HD (Promega, USA). Three days after transfection, the expression and localization of EGFP-fused proteins were examined using a FLoid™ cell imaging station (Life Technologies, USA)[17].

## Fluorescence in situ hybridization (FISH) and immunocytochemistry (ICC)

Cells seeded onto poly-lysine-coated coverslips (C1210, Matsunami, Japan) were placed in the wells of a six-well plate and fixed with 4% [v/v] paraformaldehyde in PBS (FUJIFILM, Japan) for 20 min, followed by permeabilization in 70% [v/v] ethanol for 1 h. For prehybridization, the cells were treated with hybridization buffer (10% [v/v] formamide, 5x SSC, 50 μg/ml heparin sodium salt, 0.1% [v/v] Tween 20, 100 μg/mL tRNA, and 100 μg/mL sheared boiled salmon sperm DNA) for 1 h. The cells were hybridized overnight with 0.5 μM *Wolbachia*-specific probe in hybridization buffer at room temperature. The probe was designed to bind *Wolbachia* 16 S rRNA and labeled with Texas Red [5′-Texas Red-CTTCTGTGAGTACCGTCATTATC-3′ (Fasmac, Japan)]. After overnight hybridization, the cells were washed using the following buffers: hybridization buffer, 1:1 hybridization buffer and PBS-T (0.1% [v/v] Triton X-100 in PBS), and finally PBS-T. When OfMasc-GFP was transiently expressed in the cells, FISH probe hybridization was performed before ICC staining[28]. The cells were incubated with anti-GFP antibody (598, 1:400 dilution, MBL, Japan) in ICC buffer (0.1% [w/v] BSA/PBS with 0.1% [v/v] Tween 20) for 1 h at room temperature. After 3 washes with ICC buffer, the cells were incubated with Alexa Fluor 488 F(ab')2 fragment of goat anti-rabbit IgG (H + L) (11070: 1:400 dilution, Invitrogen, USA) and DAPI (D523: 1 μg/ml, DOJINDO, Japan) for 1 h at room temperature. After 3 washes with ICC buffer, the cells were mounted with ProLong Gold (Invitrogen, USA). The stained cells were analyzed using a FLoid™ cell imaging station (Life Technologies) or a Leica STELLARIS5 confocal microscope (Leica, Germany) with an HC PL APO CS2 20× DRY objective lens (N.A. = 0.75), HC PL APO CS2 63× OIL objective lens (N.A. = 1.40), and HyD detectors. The Leica STELLARIS5 microscope was controlled by Leica Application Suite X Version 4.4.0.24861 (Leica), DAPI and Alexa Fluor 488 were excited with a 405 nm DMDO laser and a 488 nm tuned white light laser, respectively.

## Western blotting

Cells were homogenized in RIPA buffer (150 mM NaCl, 5 mM EDTA pH 8.0, 50 mM Tris-HCl pH 8.0, 1% NP-40, 0.5 % sodium deoxycholate, 0.1% SDS) and 4× SDS sample buffer (12% SDS, 25% glycerol, 150 mM Tris-HCl pH6.8, 0.05% BPB, 0.2 M DTT) (3:1). The homogenate was passed more than ten times thorough a 27-gauge needle or sonicated to shear genomic DNAs. After boiling for 5 min, the proteins were separated on 4−12% Bis-Tris gels (NuPAGE, Invitrogen, USA) in MOPS buffer using an XCell SureLock mini-cell (Invitrogen, USA). The proteins were transferred to PVDF membranes using an XCell II blot module (Invitrogen, USA) according to the manufacturer's protocol. The membranes were blocked with 4% Block Ace (DS Pharma

Biomedical, Japan), followed by incubation with primary antibody: anti-GFP antibody (598: 1:5000–1:10,000 dilution, MBL, Japan), anti-Oscar antibody [Eurofins Genomics, Japan, raised against a synthetic peptide (a.a. 1512–1525 of Oscar) 1:1000 dilution], anti-FLAG antibody (F1804: 1:3000 dilution, Sigma-Aldrich, USA) or anti-actin antibody (sc-1616-R: 1:2000 dilution, Santa Cruz, USA) in antibody dilution buffer Kiwami Setsuyaku-kun (DRC, Japan). After incubation with the primary antibody, the membrane was washed three times with TBS-T buffer, and incubated with secondary antibody (ab 98505 or ab 97032, Abcam, UK). After incubation with the secondary antibody, the membrane was washed three times with TBS-T buffer, and stained using a BCIP-NBT Solution kit for Alkaline Phosphatase Stain (Nacalai, Japan). Antibody-stained proteins were detected using a ChemiDoc XRS Plus imaging system (Bio-Rad, USA) and quantified by Image Lab software (Bio-Rad, USA). The uncropped and unprocessed scans used for figures are provided in the Source Data file.

### Production of recombinant Oscar protein
Oscar protein was produced by PUREfrex®2.0 with DnaK mix (Gene-Frontier, Japan) and concentrated using an Amicon Ultra filter unit (UFC510008, Millipore, USA). Production was verified by Western blotting using anti-Oscar antibody.

### LC-MS/MS-based identification of OfMasc-interacting proteins
OfMasc-GFP derivatives or GFP were transiently expressed in OfT1C cells ($3.2 \times 10^6$ cells) seeded in 10 cm-diameter culture dishes by transfection using FuGene HD (Promega). At 2 days post transfection, OfT1C cells were treated with 100 nM PS-341 (Bortezomib) for 1 day. The OfT1C cells were then fixed with 0.1% formaldehyde for 10 min at room temperature, and the fixation was quenched with 300 mM glycine. The cells were collected by scraping with disposable scrapers and centrifugation at 1,200 g, 4 °C for 10 min, and washed three times with chilled HEPES-saline (20 mM HEPES-KOH pH 7.5, 137 mM NaCl). The cells were then lysed on ice for 10 min in 1 mL of RIPA buffer (20 mM HEPES-KOH pH 7.5, 1 mM EGTA, 1 mM MgCl$_2$, 150 mM NaCl, 0.25% Na-deoxycholate, 0.05% SDS, 1% NP-40) supplemented with protease inhibitor cocktail cOmplete EDTA-free (Roche, Switzerland) and Benzonase (Merck, Germany). After centrifugation at 17,000 g, 4 °C for 10 min, the supernatants were incubated with GFP-Trap Magnetic Agarose (ChromoTek, Germany) for 3 h at 4 °C under gentle rotation. The magnetic agarose beads were collected using a magnetic stand, washed four times with RIPA buffer, and then twice with 50 mM ammonium bicarbonate buffer. Proteins bound to the beads were digested by adding 200 ng of Trypsin/Lys-C mix (Promega) for 16 h at 37 °C. The digests were reduced, alkylated, acidified with trifluoroacetic acid (TFA), and desalted using a GL-Tip SDB (GL Sciences, Japan). The eluates were evaporated in a SpeedVac concentrator and dissolved in 0.1% TFA and 3% acetonitrile (ACN). LC-MS/MS analysis of the resultant peptides was performed using an EASY-nLC 1200 UHPLC connected to an Orbitrap Fusion mass spectrometer equipped with a nanoelectrospray ion source (Thermo Fisher Scientific). The peptides were separated on a 75 μm inner diameter × 150 mm C18 reversed-phase column (Nikkyo Technos, Japan) with a linear 4–32% ACN gradient for 0–100 min followed by an increase to 80% ACN for 10 min. The mass spectrometer was operated in a data-dependent acquisition mode with a maximum duty cycle of 3 s. MS1 spectra were measured with a resolution of 120,000, an automatic gain control (AGC) target of $4 \times 10^5$, and a mass range from 375 to 1,500 m/z. Higher-energy collisional dissociation (HCD) MS/MS spectra were acquired in the linear ion trap with an AGC target of $1 \times 10^4$, an isolation window of 1.6 m/z, a maximum injection time of 35 ms, and a normalized collision energy of 30. Dynamic exclusion was set to 20 s. Raw data were directly analyzed against the wFur encoded protein data predicted from an in-house assembled wFur draft genomic sequence and *O. furnacalis* protein data (GCF_004193835.1_ASM419383v1_protein.faa) downloaded from NCBI supplemented with OfMasc-GFP and GFP-Trap sequences using Proteome Discoverer version 2.4 (Thermo Fisher Scientific) with Sequest HT search engine. The search parameters were as follows: (a) trypsin as an enzyme with up to two missed cleavages; (b) precursor mass tolerance of 10 ppm; (c) fragment mass tolerance of 0.6 Da; (d) carbamidomethylation of cysteine as a fixed modification; and (e) acetylation of protein N-terminus and oxidation of methionine as variable modifications. Peptides and proteins were filtered at a false discovery rate (FDR) of 1% using the percolator node and the protein FDR validator node, respectively. Label-free precursor ion quantification was performed using the precursor ions quantifier node, and normalization was performed such that the total sum of abundance values for each sample over all peptides was the same.

Several selected peptides of Oscar and GFP were measured by PRM[29], an MS/MS-based targeted quantification method using high-resolution MS. Targeted HCD MS/MS scans were acquired by a time-scheduled inclusion list at a resolution of 60,000, an AGC target of $1.5 \times 10^5$, an isolation window of 1.6 m/z, a maximum injection time of 500 ms, and a normalized collision energy of 30. Time alignment and relative quantification of the transitions were performed using Skyline software.

### Immunoprecipitation of Oscar-bound OfMasc-GFP protein
OfT1C/tet cells ($4 \times 10^6$ cells in 10 cm-diameter culture dishes) were co-transfected with *3×FLAG-Oscar* and *GFP*, *OfMasc-GFP* or *Del1-OfMasc-GFP* using FuGene HD. At 3 days post transfection, the cells were treated with 200 nM PS-341 for 6 h, fixed with 0.1% formaldehyde for 10 min at room temperature, and the fixation was quenched with 300 mM glycine. The cells were collected by scraping with disposable scrapers and centrifugation at 1200 g, 4 °C for 10 min. The cells were then lysed on ice for 10 min in 1 mL of RIPA-EDTA buffer (20 mM HEPES-KOH pH 7.5, 1 mM EDTA, 1 mM MgCl$_2$, 150 mM NaCl, 0.25% Na-deoxycholate, 0.05% SDS, 1% NP-40) supplemented with protease inhibitor cocktail cOmplete EDTA-free and Benzonase. After centrifugation at 17,000 g, 4 °C for 10 min, the supernatants were collected and protein concentration was determined using Pierce™ BCA Protein Assay Kit (Thermo Fisher Scientific, USA). The equal amounts of input proteins were incubated with anti-FLAG M2 magnetic beads (Sigma-Aldrich, USA) for 3 h at 4 °C under gentle rotation. The magnetic agarose beads were collected using a magnetic stand, washed three times with RIPA-EDTA buffer, and then mixed with 2× SDS sample buffer without 0.2 M DTT to elute binding proteins. Immediately before electrophoresis, loading samples were added with 2× SDS sample buffer containing 0.2 M DTT. Western blotting was performed as described above with Monoclonal ANTI-FLAG® M2 antibody produced in mouse (F1804: 1:3000 dilution, Sigma-Aldrich, USA) and anti-GFP antibody (598: 1:5000–1:10,000 dilution, MBL) as primary antibodies and Goat anti-Mouse IgG (H + L) Secondary Antibody, HRP (626520: 1:5000, Thermo Fisher Scientific, USA) and Peroxidase AffiniPure Goat Anti-Rabbit IgG (H + L)(111-035-144: 1:10,000, Jackson Immuno Research Laboratories, Inc., USA) as secondary antibodies. Signals were detected with ECL™ Prime Western Blotting Detection Reagent (RPN2232, Cytiva, Japan) using ChemiDoc XRS Plus imaging system. Signal intensities were measured using Image Lab software. The uncropped and unprocessed scans used for figures are provided in the Source Data file.

### Genome assembly and synteny analysis
To obtain genome sequences of wFur, an ovary of a *Wolbachia*-infected female was sampled. Genomic DNA was purified using QIAGEN

Genomic-tip 100/G according to manufacturer's protocol, and was sequenced by Illumina Novaseq 6000 (150 bp paired-end) platforms for short-read sequencing and PacBio Sequel I for long-read sequencing. The long-reads that were obtained were assembled by Canu v2.1[30]. An estimated genome size of 438.5 Mb was given as a parameter with reference to the existing genome assembly of *O. furnacalis* (GCA_004193835.1). Every generated contig was subject to BLASTn (v.2.7.1) search as a query against Refseq representative prokaryotic genomes database (ref_prok_rep_genomes [https://ftp.ncbi.nlm.nih.gov/blast/db/]). Contigs that were aligned to publicly available *Wolbachia* sequences were identified as *w*Fur-derived sequences. BWA v0.7.17[31] was used to map Illumina short-reads to the candidate *w*Fur draft genome with BWA-MEM mode, and Pilon v1.23[32] was utilized to polish the assembly. Stand-alone PGAP v.2021-05-19.build5429[33] was utilized to annotate the polished *w*Fur genome.

The gene arrangement of the *Oscar* locus in the *w*Fur genome was compared to that of the *w*Pip genome. BLASTp (v.2.9.0) searches were conducted between translated protein sequences of *w*Fur (generated as PGAP output) and *w*Pip (AM999887) in both directions, and genes of reciprocal best hits were regarded as orthologs. In addition, for *Oscar*, a homologous region in the *w*Pip genome was explored using BLASTn (v.2.9.0).

### Phylogenetic analysis of *w*Fur *wmk* and *wmk-like* genes
Homologs of *wmk* in the *w*Fur genome were identified using BLASTn (v.2.12.0) and tBLASTn (v.2.12.0) with an E-value cutoff of 1e-10. The nucleotide and amino acid sequences of *w*Mel *wmk* (locus tag WD0626 in the *w*Mel genome sequence AE017196.1) were used as queries for BLASTn (v.2.12.0) and tBLASTn (v.2.12.0), respectively. The loci annotated as pseudogenes by the PGAP annotation were excluded from the subsequent phylogenetic analysis. The deduced amino acid sequences of *wmk* and its homologs in the *w*Fur and *w*Mel genomes were aligned using ClustalW v.2.1 with default parameters. A maximum likelihood tree was constructed using IQ-TREE v.2.2.0.3[34] with 1000 standard bootstrap replicates. The best-fitting substitution model was estimated by ModelFinder, which is implemented in IQ-TREE, and Q.mammal+F + G4 was selected according to Bayesian information criterion.

### Protein structure prediction by AlphaFold2
Domain architectures of Oscar were analyzed using SMART (a Simple Modular Architecture Research Tool). Protein structures were predicted using an in-house Linux computer in which AlphaFold ver.2.0 was installed[35]. AlphaFold2 was installed by Docker and CUDA Toolkit 11.1. full_dbs was used as a database and max_template_date was defined as 2021-07-14. For prediction of the protein complex, two protein sequences were connected with a U80 linker and then subjected to AlphaFold2 prediction. Due to limitation of our computer, only a single model was predicted and structure optimization was not performed when the entry data of protein sequences was large. The heatmaps of predicted aligned errors were made using the pae2png.py script (https://github.com/CYP152N1/plddt2csv).

### Functional analysis of *Oscar* in *B. mori* and *O. furnacalis* embryos
The DNA template for cRNA synthesis was amplified by PCR using plasmid DNA containing a codon-optimized *Oscar* fragment (*Oscar*-pIZ/V5-His-g3) or *GFP* (control) cDNA[6]. Primers used for PCR are listed below:

pIZ-F-T7:  5′-TAATACGACTCACTATAGGGAGACAGTTGAACAGCATCTGTTC-3′
pIZ-R: 5′-GACAATACAAACTAAGATTTAGTCAG-3′
Oscar-cRNA-F-T7:
5′-TAATACGACTCACTATAGGGAGAATGGAGGATAGACATATCCCGTTCC-3′
Oscar-cRNA-R: 5′-TTATCTGCCGCCTTTTCCTTTGGAA-3′

Capped, poly(A)-tailed cRNA was synthesized using an mMES-SAGE mMACHINE T7 Ultra Kit according to the manufacturer's protocol (Ambion, USA). *Oscar* or *GFP* cRNA solution (200 ng/μL in 100 mM potassium acetate, 2 mM magnesium acetate, 30 mM HEPES-KOH; pH7.4) was injected into *B. mori* embryos within 2–3 h after oviposition[11,36]. The injected embryos were collected at 48 h after injection and total RNA and genomic DNA was prepared from a single embryo[11,36]. Total RNA was subjected to RT-PCR experiments for *Bmdsx* and genomic DNA was used for molecular sexing by PCR using a W chromosome specific primer, *Musashi*[11,36]. The hatched larvae and unhatched developed embryos were also collected and subjected to molecular sexing PCR. Injection of cRNA into *O. furnacalis* embryos was performed according to the methods in *B. mori* with some optimization. The egg masses, all embryos of which were injected with *GFP* or *Oscar* cRNA, were collected at 1 or 2 days post-injection and subjected to RT-PCR experiments for *Ofdsx*. The hatched larvae were also collected and subjected to molecular sexing by qPCR using genomic DNA of the Z-linked gene *Kettin* and autosomal gene *EF-1α* as standards[6]. The adult *O. furnacalis* moths emerged from *GFP* or *Oscar* cRNA-injected embryos were sexed based on the external morphology.

### Statistics
Statistical analyses were performed using Prism 9 software (GraphPad, USA).

### Reporting summary
Further information on research design is available in the Nature Portfolio Reporting Summary linked to this article.

### Data availability
Source data are provided with this paper. The *Oscar* coding sequence, the 40.5 kbp-long *w*Fur genome sequence around the *Oscar* locus, and *w*Fur genome have been deposited in GenBank under accession numbers LC657087, LC659497, and CP096925, respectively. *SfMasc*, *SfdsxM*, *SfdsxF1*, and *SfdsxF2* have been deposited in GenBank under accession numbers LC716474, LC716647, LC716648, and LC716649, respectively. The MS proteomics data have been deposited in the ProteomeXchange Consortium via the jPOST partner repository with the data set identifier PXD034837. All other data are included as Supplementary Information and Supplementary Data. The public data used are as follows: *O. furnacalis* protein data (GCF_004193835.1_ASM419383v1_protein.faa), genome assembly of *O. furnacalis* (GCA_004193835.1), Refseq representative prokaryotic genomes database (ref_prok_rep_genomes, [https://ftp.ncbi.nlm.nih.gov/blast/db/]), translated protein sequences of *w*Pip (AM999887), and the *w*Mel genome sequence (AE017196.1). Source data are provided with this paper.

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

## Acknowledgements

We thank S.G. Kamita for critical reading of the manuscript, M. Ote for providing the FISH protocol and critical reading of the manuscript, M. Kawano for LC-MS/MS analysis, N. Matsuda for critical comments and technical suggestions, N. Nakashima for insect rearing and sampling, M. Kawamoto, H. Hikida, and Y. Suzuki for their contribution of the initial stage of the project, the Technology Advancement Center, Graduate School of Agricultural and Life Sciences, The University of Tokyo for TEM analysis, the Institute for Sustainable Agro-ecosystem Services, The University of Tokyo, for facilitating the mulberry cultivation, the Biotron Facility at The University of Tokyo for insect rearing, and C. Yasunaga-Aoki and R. Sato for providing BmN-4 and Sf-9 cell lines, respectively. This work was supported by Grants-in-Aid for Scientific Research on Innovative Areas "Spectrum of the Sex: a continuity of phenotypes between female and male" (17H06431) to S.K. and T.K., Grant-in-Aid for Scientific Research (A) (22H00366) to S.K., Grant-in-Aid for Challenging Exploratory Research (15K14893) to S.K., JST SPRING (JPMJSP2108) to K.H.

## Author contributions

S.K. and T.K. initiated the project. S.K., N.M.-I., K.H., T.Fukui, and T.K. conceived and designed the experiments. T.K. generated *O. furnacalis* cell lines. S.K. characterized *O. furnacalis* cell lines and established the cell-based assay system. N.M.-I., S.K., K.H., T.Fukui, T.K., and T.M. performed molecular biological experiments. N.M.-I. and K.H. performed biochemical experiments. K.N. and H.K. performed LC-MS/MS analysis. K.H., N.M.-I., K.N., and H.K. identified Oscar. S.K., N.M.-I., and K.H. performed functional characterization of *Oscar* in cultured cells. S.K. and T.M. characterized *w*Fur *wmk* genes. T.K. performed functional analysis of *Oscar* in *B. mori* embryos. T.Fukui performed functional analysis of *Oscar* in *O. furnacalis* embryos. T.M. performed *w*Fur genome analysis. K.S. performed AlphaFold2 prediction. N.M.-I., K.H., H.T., and S.-i.A. performed confocal microscopy experiments. T.Fujii performed sex pheromone analysis. S.K. wrote the manuscript with intellectual input from all authors. S.K. supervised the project.

## Competing interests

The authors declare no competing interests.
