## [Peer Review File · Nature Communications]

A Wolbachia factor for male killing in lepidopteran insectsReviewer #1 (Remarks to the Author):

In this study, the authors uncovered the long-sought molecular mechanism of Wolbachia-induced male-killing in insects of the order Lepidoptera. In cell lines derived from the ovaries of the Wolbachia-infected moth *Ostrinia furnacalis*, they identified a protein called Oscar, through which Wolbachia inhibits the Masculinizer (Masc) protein which is required for male development and also for dosage compensation. Using a detailed analysis of the structure of Oscar and a series of sophisticated experiments, they clearly demonstrated that it interacts with the Masc protein, inhibiting masculinization controlled by Masc. In addition, they identified the Masc domain through which Oscar interacts with Masc and demonstrated it experimentally. The authors also demonstrated a similar function of Oscar in cell lines of four other Lepidoptera species, including the genetic model of Lepidoptera, the silkworm *Bombyx mori*, and also in laboratory strains and field-collected specimens of two species. Overall, this is a great discovery that deserves to be published in this highly respected journal.

The manuscript is perfectly written and the results are very well documented with illustrations. It was a great pleasure for me to review such a well-done study.

Specific comment

Discussion: in Fig. 4f, the authors have proposed a model for Oscar-mediated male killing in *Ostrinia furnacalis* that explains the mechanism of Wolbachia-induced male killing and that probably applies to other lepidopteran species. However, previous studies on the interaction between Wolbachia and *Ostrinia* have shown that *Ostrinia* male-killing strains cured with tetracycline produce all-male offspring, i.e. female offspring die (e.g. Sugimoto and Ishikawa 2012), suggesting a feminizing function of Wolbachia. Of course, this function is outside your model. Do you have a hypothesis on how this might work? If so, it would be interesting to briefly mention it in the Discussion.

Minor suggestions

L237: ... indicating that more than 11 copies ...

L318-320: except for the first word, use lowercase letters in the general words of the title of Arai et al. (2019)

L360-361: except for the first word, use lowercase letters in the general words of the title of Liu et al. (2019)

L370, L585, L841, L937, L957: EF-1 α

Fig. 2e: the two cysteine residues of OfMasc, shown in magenta on the top two panels, may be difficult to see. It might be useful to mark them with arrows or asterisks.

L939 and L948: t-test [*t*] should be written in italics]

L1006-1008, Extended Data Fig. 6c, legend: it would be worth mentioning the sex of examined specimens of the lab strains as well as field-collected *Ostrinia* species.

Frantisek Marec

26 May 2022

Reviewer #2 (Remarks to the Author):

Katsuma and colleagues report convincing data that a single key Wolbachia (wFur) factor is responsible for driving male killing in lepidopterans. At this point this is probably the most complete story regarding a mechanism for one of the reproductive manipulations by Wolbachia, although there is still some way to go for a molecular understanding.

The same group had previously identified the protein Masculinizer (Masc) of the moth *Ostrinia furnacalis* as the target for wFur-induced male killing through interference with Masc's function in dosage compensation. Here they set up cell lines derived from wFur-infected *O. furnacalis* embryos showed the female splicing pattern for doublesex (*dsx*), and Masc cRNA transfection caused production of male-specific *dsx*(M) mRNA. In GFP-Masc transfectants of wFur-containing cells, very little GFP signal was seen, unlike Wolbachia-cured cells. Proteasome inhibitors increased the GFP signal, and this was also seen by Western blotting. An N-terminal zinc finger, ZF1, of Masc was found to be important for the response to wFur, suggesting a potential interaction with a Wolbachia factor that leads to Masc degradation. Using LC-MS/MS to search for N-terminal domain-dependent Wolbachia interactors, the authors identified a protein they call Oscar (which I hope will not be confused with the insect embryonic developmental factor Oskar). Curiously, binding was not validated by directed pulldown assays, although it seems likely.

Oscar is a 201 kDa protein with 40 ankyrin repeats followed by a CidB-related deubiquitylase domain. It is interesting that the *D. melanogaster* male-killing factor, called Spaid, from the bacterium *Spiroplasma poulsonii* also has a (smaller) set of ankyrin repeats followed by a deubiquitylase domain. The authors went on to show that transfection of Oscar RNA into the wFur-infected cultured cells inhibits Masc accumulation and *dsx*(M) mRNA expression, and this was not observed if Masc lacked the N-terminal domain required for Oscar binding. *B. mori* and *O. furnacalis* embryos injected with Oskar cRNA did not produce male-specific splicing products, and all the viable progeny were female (and the dead embryos were almost all male). These results make a convincing case that Oscar is the wFur male-killing factor.

I have a very positive opinion of this study and think it should ultimately be published in Nature Communications. I would first like to see the follow issues addressed:

Major:

1. The authors have a nice cell culture system to evaluate Oscar mutants. No analysis is done for the Ulp1-like domain. All the active residues of the protease appear to be present. It would be straightforward to test at least in cultured cells whether an active-site mutant affects Masc accumulation and *dsx*(M) expression. It would also be nice to include ubiquitin-AMC enzyme assays for this domain, but I would not demand this.
2. It would be useful to see a co-IP or pulldown analysis of Masc and Ndel-Masc with Oscar, at least from the cultured cells. This would at least be suggestive of direct interaction.
3. The AlphaFold2 analysis is okay, but it could also use validation. If the authors are confident of these structures, they could create mutants that should block binding or use directed crosslinking to validate their structures.

Minor:

1. Both Ref. 3 and 16 should be included for the discovery of the CifAB factors noted on line 60, p. 3.
2. The authors several times state from proteasome inhibition that degradation by the ubiquitin-proteasome system is involved. However, no test of a ubiquitin requirement is done, and there are protein substrates that do not require it (although I doubt this is true here). This could just be written more carefully, or ubiquitin pathway inhibition could be tested using commercially available E1 ubiquitin activating-enzyme inhibitors (I think Takeda).
3. The authors note that a number of ubiquitin pathway enzymes have ankyrin repeats. Do they believe that Oscar is part of a ubiquitin ligase (E3) complex? This would make sense with respect to the effects on Masc levels. The deubiquitylase could have an editing function or could regulate the E3 activity. If the authors do an Oscar pulldown/MS analysis, do they identify other components of ubiquitin ligase complexes or pathways?
4. Although it is probably not critical to the current paper, I did not understand the evidence that

Masc controls masculinization in the cytoplasm and dosage compensation in the nucleus. This part could be written more clearly.

5. On line 241, the BmIMP(M) factor is mentioned but I'm not sure this was introduced earlier. Not sure where it fits in exactly.

Reviewer #3 (Remarks to the Author):

Dear authors,

The manuscript by Katsuma et al. identifies and characterizes a novel Wolbachia factor involved in male-killing in *Ostrinia*. The authors established new cell-lines with and without Wolbachia and showed that these cell-lines are phenocopies of infected embryos. Using GFP-fused Masc they showed that accumulated Masc is inhibited in Wolbachia-infected cells but not in uninfected cells. The interacting parts of Masc were tested with different induced deletions and the first zinc finger was shown to be involved in the binding to the Wolbachia factor. The Wolbachia factor was identified using a clever combination of the full-length GFP-Masc and the deleted GFP-Masc in an immunoprecipitation assay. The authors show that part of the novel Wolbachia protein, Oscar, contains a CifB terminus-like part which was shown to be involved in another form of reproductive manipulation in other insects. The authors present convincing data that indeed a male-killing factor has been functionally characterized for the first time. The evolution of this factor touched upon in the discussion but more phylogenetic analysis would be necessary to solidify this story. In our view the authors have shown an impressive amount of work that is of tremendous interest to the scientific community.

After revision (among which putting the work here in a more extended framework of previous research, improving the details in the materials and methods), and the addition of missing data (wFur genome) we think this manuscript is publishable.

Kind wishes,
Dr. Eveline Verhulst
Dr. Sander Visser

Structural suggestions

Abstract

I would suggest the authors to remove the references in the abstract as this is unusual. Please add titles "Introduction" and "Results" to the manuscript.

Specific comments to the text

Line 56-65: This section does not entirely do justice to the contents of Perlmutter et al. Especially the formulation in lines 64-67 suggest that wMel induces male-killing in *D. melanogaster*, which is not the case. Perlmutter and all indicate that they studied the wMel strain to find male-killing genes as they found that the 99,7% similar wRec strain can induce both male-killing and CI in different *Drosophila* species. The identified wmk gene has homologs in many male-killing Wolbachia strains as have been cifA and cifB, and are not just residing on wMel.

This line needs to be reformulated to indicate that wmk is a candidate for the male-killing phenotype of many Wolbachia strains.

Also in the discussion the findings in this manuscript should be put much more into context of what was found in the study of Perlmutter et al. Is Oscar completely different from wmk? Does wFur also contain wmk if Oscar is not the same as wmk?

We suggest the authors use this paper more in the discussion, and adjust the mentioned section in the introduction.

Line 69: suggestion to change "male-killing" to infected

Line 81: please replace "from" by "in".

Line 87: The reference to the figure in the context of this sentence is unclear to me. The importance and relevance of Fig 1a is also unclear in the context here. In addition the pictures are hard to interpret and not explained well in the caption. Are the black dots Wolbachia?

Lines 96-97 and line 102-104: The suggestion that Masc proteins have two roles (masculinization and dosage compensation) depending on their localization is not shown in this manuscript, but can also not be concluded (and has not been concluded) in the referenced article. Therefore, the writing here that the masculinizing function of Masc protein occurs in the cytoplasm is not valid and should be adjusted, or additional references should be added that can support this statement.

In general the logics of this paragraph (line 96-107) are hard to follow for researchers not intimately known with the details of Ostrinia previous work. It would be good to clarify that overexpression of OfMasc with cRNA was shown before to prevent male-killing but that it was not enough to masculinize embryos. The description of it is now a bit cryptic.

Line 99: please replace "survived" to "surviving".

Line 111 (Fig. 1b): What are the readers supposed to see in these two pictures? Are these pictures necessary for the main figure? We think they are better suited for the supplementary.

Lines 114-115: default splicing is not shown in Figure 1e to which is referred here. Figure 1e shows splicing of Ofdsx after transfection with GFP and OfMasc-GFP.

Line 116: transfection with only OfMasc is not shown in Figure 1E, do the authors mean GFP here instead?

Please explain the cDNA here. What does it encode? Cannot find the description of these GFP-OfMasc fused protein in the material and methods either.

Line 118-119: Please explain why OfMasc expression did not affect the splicing pattern of Ofdsx. Is this again due to the lower levels of OfMasc not meeting the threshold necessary to induce the shift in splicing?

Supplementary Fig. 1b is the same as Fig. 1e. Please check and adjust.

Lines 122-125: please change "origin is same" to "origin is the same".

Line 148: please replace "for" by "in".

Lines 152-155: It is unclear from the description how the final candidate protein was identified from the 1045 initial proteins. Was there only a single protein derived from wFur recovered among the 1045 candidates and the additional 1044 proteins all originated from the host? The way this is currently written, it seems that there was only one wFur protein that interacted with the OfMasc N-terminal region suggesting there were more wFur proteins that were recovered but did not interact with the N-terminal which is confusing because it would require additional experiments to determine which region of the OfMasc protein is bound. Please clarify.

Line 194: It is difficult to see in the figure that OfMasc changes configuration. It can be seen that Oscar changes configuration, but OfMasc change in configuration is difficult. Is it possible to have a separated image of the configuration difference of OfMasc when captured by Oscar?

Line 196-199: This part of the paragraph has no links to the rest of the paragraph and its message is unclear. Please expand on this or move this section to a more appropriate location.

Lines 204-206: Rather than a partial deletion in an ancestral Wolbachia of wPip, this could have just as well been an insertion/fusion in an ancestral lepidopteran Wolbachia lineage. A phylogeny of the different Wolbachia strains with the WO prophage would be helpful to put this in better context.

Lines 230-232: I would recommend to rephrase this sentence a little as the expression level of *dsx* is likely not affected by Oscar, but the splicing of *dsx* is (indirectly) shifted from male- to female-specific. Therefore, stating that Oscar reduces the expression level of the male-type *dsx* variant is not fully accurate.

Line 240: please change the second "expression" for "splicing".

Line 245: this statement is too strong. The *dsx* splicing data from the BmN-4 cells is showing small effects for some of the tested Masc proteins. More importantly, given the presented data, Oscar is able to reduce Masc accumulation, but no data is presented on the presence of Oscar in other Wolbachia species/strains in Lepidoptera, so stating that this is "strongly suggesting that Oscar commonly inhibits" this pathway is way too strong. Please tone down the statement to match the observed results.

Line 250: the use of expression here is confusing. We suggest to change these instances to splicing in the paragraph.

Line 250-252: Were the hatched larvae injected with Oscar cRNA genetically or phenotypically female? Please add in the text.

In Fig. 4b: Why are the sample sizes for Oscar females and GFP males much lower than the other two groups?

In Fig. 4e: Why not indicate the male band as well? Indicating the female band is a bit odd here, as it's equally as important to also know what the other band is.

In addition, are all the dead embryos shown in this picture? Meaning that only 6 died in the GFP group, but 16 died in the Oscar group? Please add this to the caption.

Lines 269-271 and Fig. 4f: I would strongly recommend to remove the cytoplasm and nucleus sections, as this was not specifically shown in this study nor in the cited studies.

Lines 272-273: similar to the previous comment, this study shows that Masc is targeted by Oscar, but not that Oscar specifically affects dosage compensation. This needs to be elaborated upon and needs a reference, or, the statement needs to be toned down.

Line 284: We don't agree that 2 cases of proteins with ankyrin repeats suggests that this is a common pathway through which sexual manipulation by symbiotic bacteria is established. This statement is worded too strongly, given the available data. Please tone it down.

One point we are missing in the discussion is based on previous data published by the authors on the effect of wFur on the expression of OfMasc. In Fukui et al. 2015 (doi: 10.1371/journal.ppat.1005048), the authors found and confirmed that OfMasc expression is lower in Wolbachia-infected individuals than in non-infected individuals (see Figure 3 of that article). Given that Oscar interacts with OfMasc (and Masc of other lepidopteran species) on the protein level rather than the DNA or RNA level, we would like to ask the authors to discuss how the previous data fit with the current data in terms of expression levels.

In addition, as mentioned above, please put the finding in this manuscript in the context of the *wmk* gene found in Perlmutter et al.

Comments to Methods

In general

Please make sure to include the countries for all companies, and not only for few companies.

Please make sure to remove the sequences from the material and methods, submit them to GenBank and add accession numbers in their place in the text.

Line 460: SFM stands for Serum Free Medium, so the use of the word "medium" behind SFM is redundant and should be removed. This is a repeated error, so please correct it throughout the manuscript.

Lines 486-578: The details of the experiment are not described in this subsection to the extent that the experiments can be replicated. The starting concentration of RNA for cDNA conversion can make a difference in the final results and should be given. In addition, the conditions for the RT-PCR are not given, i.e. was a 2- or 3-step PCR cycle used (and if it was a 3-step, what annealing temperature was used?), the amount/volume of template cDNA is not given, the total reaction volumes are not given. The information given here is the same as in the four referred articles, therefore, this information can also not be found in those articles and should be included here.

Line 495: Are these sequences that are predicted bioinformatically, or are these sequenced PCR fragments? Please describe that the fragments were sequenced (and the steps preceding this) if they were.

Lines 496-568: These sequences don't belong in the Methods section and should rather be submitted to GenBank. Instead, the GenBank identifiers can be given here in the text.

Line 585: No primer sequences or reference given for the EF1a primers.

Line 789: please correct to "a Wolbachia-infected....."

Line 792: details about the Illumina sequencing data are missing, i.e. specific Illumina platform, read size, insert size, single- or paired-end. Also, was there any software used for quality control of either the long or short read datasets before genome assembly and read mapping?

In addition, where can the sequence of wFur data be found? This should be uploaded to GenBank and the identifier can be given here.

Paragraph starting at line 819: Details regarding the PCR are missing, apart from the primers. In addition, there are several steps missing starting at line 832, i.e. RNA/DNA isolation, cDNA conversion, etc.

Mentioned in line 833 that embryos were subjected to RT-PCR or molecular sexing, is this correct or were embryos used for both these experiments? In the figures of the manuscript it appears that the latter should be true. In that case, there is a description missing of DNA and RNA isolation of single embryos.

Comments to figures

Fig. 1.

1a-b: I recommend to change the colour of "a" and "b" to black, or, if it improves visibility, add a white box around the black letter. The white font does not improve the visibility.

Please also keep the placing of the scale bar consistent for all pictures, and add it in all pictures.

1d: please correct the scale bar, the original is also visible in the bottom left corner of each picture.

1f: We recommend putting the same style thick scale bar on each picture because the narrow version is not properly visible (in addition the writing of the thin scale bar is still visible in the first picture, I would correct that).

Fig. 2.

2a: Similar issue with the scale bar as in 1f. In addition, the ends of the "original" scale bar are visible above the added scale bar in the first picture, revealing that the added scale bar is not exactly the same size as the original. This definitely should be fixed for all scale bars in all figures (it is for example also visible in figure 3a).

2e: The scales of the heatmap are not readable and axes have no description. Add a better description of these heat maps to the caption.

Fig. 3.

3a: Different pictures not aligned properly. Please fix the scale bars here as well.

Extended Data Fig. 8d: We are surprised to see that in none of the controls there is any variation between the 3 replicates, especially considering the data in Extended Data Fig. 8b. Therefore, I question how the controls were set to 100% (and how is the undoubtedly present variation between the 3 replicates incorporated)?. Is there any explanation why transfection with the different Masc-GFPs results in variation of BmIMPM splicing levels, but co-transfection with an empty vector, nullifies this variation. Was transfection done three times independently, or did the 3 samples originate from a single transfection experiment?

Point-by-point responses to the reviewer's comments

We thank three reviewers for their valuable comments regarding our manuscript NCOMMS-22-18013-T.

Reviewer #1 (Remarks to the Author):

In this study, the authors uncovered the long-sought molecular mechanism of Wolbachia-induced male-killing in insects of the order Lepidoptera. In cell lines derived from the ovaries of the Wolbachia-infected moth *Ostrinia furnacalis*, they identified a protein called Oscar, through which Wolbachia inhibits the Maculinizer (Masc) protein which is required for male development and also for dosage compensation. Using a detailed analysis of the structure of Oscar and a series of sophisticated experiments, they clearly demonstrated that it interacts with the Masc protein, inhibiting masculinization controlled by Masc. In addition, they identified the Masc domain through which Oscar interacts with Masc and demonstrated it experimentally. The authors also demonstrated a similar function of Oscar in cell lines of four other Lepidoptera species, including the genetic model of Lepidoptera, the silkworm *Bombyx mori*, and also in laboratory strains and field-collected specimens of two species.

Overall, this is a great discovery that deserves to be published in this highly respected journal.

The manuscript is perfectly written and the results are very well documented with illustrations. It was a great pleasure for me to review such a well-done study.

(Authors' reply)

We thank Prof. Marec for his encouraging comments.

Specific comment

Discussion: in Fig. 4f, the authors have proposed a model for Oscar-mediated male killing in *Ostrinia furnacalis* that explains the mechanism of Wolbachia-induced male killing and that probably applies to other lepidopteran species. However, previous studies on the interaction between Wolbachia and *Ostrinia* have shown that *Ostrinia* male-killing strains cured with tetracycline produce all-male offspring, i.e. female offspring die (e.g. Sugimoto and Ishikawa 2012), suggesting a feminizing function of Wolbachia. Of course, this function is outside your model. Do you have a hypothesis on how this might work? If so, it would be interesting to briefly mention it in the Discussion.

(Authors' reply)

Thank you for your suggestion. We think Oscar is the *Wolbachia*'s feminizing factor (i.e. inhibitor of Masc) and an endogenous *Ostrinia* feminizing factor is disrupted or inactivated in male-killing *Wolbachia*-infected *Ostrinia* strains. We added a brief hypothesis and future perspective in the Discussion section.

Minor suggestions

L237: ... indicating that more than 11 copies ...

L318-320: except for the first word, use lowercase letters in the general words of the title of Arai et al. (2019)

L360-361: except for the first word, use lowercase letters in the general words of the title of Liu et al. (2019)

(Authors' reply)

Corrected.

L370, L585, L841, L937, L957: EF-1 α

(Authors' reply)

Corrected. We also corrected the titles of y-axis of Fig. 1b, Supplementary Fig. 1g, and Supplementary Fig. 2c.

Fig. 2e: the two cysteine residues of OfMasc, shown in magenta on the top two panels, may be difficult to see. It might be useful to mark them with arrows or asterisks.

(Authors' reply)

We added magenta arrows in Fig. 2e and Supplementary Fig. 5d.

L939 and L948: t-test ["t" should be written in italics]

(Authors' reply)

Corrected.

L1006-1008, Extended Data Fig. 6c, legend: it would be worth mentioning the sex of examined specimens of the lab strains as well as field-collected *Ostrinia* species.

(Authors' reply)

We used female moths for this analysis. We added this to the legend.

Frantisek Marec

26 May 2022

(Authors' reply)

Thank you again for your helpful suggestions.

Reviewer #2 (Remarks to the Author):

Katsuma and colleagues report convincing data that a single key Wolbachia (wFur) factor is responsible for driving male killing in lepidopterans. At this point this is probably the most complete story regarding a mechanism for one of the reproductive manipulations by Wolbachia, although there is still some way to go for a molecular understanding.

The same group had previously identified the protein Masculinizer (Masc) of the moth *Ostrinia furnacalis* as the target for wFur-induced male killing through interference with Masc's function in dosage compensation. Here they set up cell lines derived from wFur-infected *O. furnacalis* embryos showed the female splicing pattern for doublesex (dsx), and Masc cRNA transfection caused production of male-specific dsx(M) mRNA. In GFP-Masc transfectants of wFur-containing cells, very little GFP signal was seen, unlike Wolbachia-cured cells. Proteasome inhibitors increased the GFP signal, and this was also seen by Western blotting. An N-terminal zinc finger, ZF1, of Masc was found to be important for the response to wFur, suggesting a potential interaction with a Wolbachia factor that leads to Masc degradation. Using LC-MS/MS to search for N-terminal domain-dependent Wolbachia interactors, the authors identified a protein they call Oscar (which I hope will not be confused with the insect embryonic developmental factor Oskar). Curiously, binding was not validated by directed pulldown assays, although it seems likely.

Oscar is a 201 kDa protein with 40 ankyrin repeats followed by a CidB-related deubiquitylase domain. It is interesting that the *D. melanogaster* male-killing factor, called Spaid, from the bacterium *Spiroplasma poulsonii* also has a (smaller) set of ankyrin repeats followed by a deubiquitylase domain. The authors went on to show that transfection of Oscar RNA into the wFur-infected cultured cells inhibits Masc accumulation and dsx(M) mRNA expression, and this was not observed if Masc lacked the N-terminal domain required for Oscar binding. *B. mori* and *O. furnacalis* embryos injected with Oskar cRNA did not produce male-specific splicing products, and all the viable progeny were female (and the dead embryos were almost all male). These results make a convincing case that Oscar is the wFur male-killing factor.

I have a very positive opinion of this study and think it should ultimately be published in Nature Communications. I would first like to see the follow issues addressed:

(Authors' reply)

We thank this Reviewer for his/her encouraging comments. As suggested, the results of co-IP experiments were added to the revised manuscript (Supplementary Fig. 3f, described below).

Major:

1. The authors have a nice cell culture system to evaluate Oscar mutants. No analysis is done for the Ulp1-like domain. All the active residues of the protease appear to be present. It would be straightforward to test at least in cultured cells whether an active-site mutant affects Masc accumulation and *dsx(M)* expression. It would also be nice to include ubiquitin-AMC enzyme assays for this domain, but I would not demand this.

(Authors' reply)

To examine the role of Oscar's CifB-like domain, we generated additional constructs for Oscar derivatives lacking CifB-like domain (DelC) or both linker and CifB-like domains (DelD). As shown in Supplementary Fig. 8b–d of the revised manuscript, CifB-like domain is slightly involved in but not essential for inhibitory activity of Masc accumulation and Masc-induced masculinization. Additional experimental results using other mutants strongly suggest that ankyrin repeats possess a crucial role for inhibition of Masc's functions (Supplementary Fig. 8b–d). Future studies will explore the role of the enzymatic activity of CifB-like domain in Oscar-induced male killing.

2. It would be useful to see a co-IP or pulldown analysis of Masc and Ndel-Masc with Oscar, at least from the cultured cells. This would at least be suggestive of direct interaction.

(Authors' reply)

As pointed out by this Reviewer, we performed co-IP experiments using OfT1C/tet cells co-transfected with *3×FLAG-Oscar* and *OfMasc-GFP* or *Del1-OfMasc-GFP* cDNA. Although *OfMasc-GFP* accumulated extremely lower compared to *Del1-OfMasc-GFP* due to proteasome-mediated degradation, relative amount of Oscar-bound *OfMasc* proteins (IPed/Input) verified that Oscar interacts with *OfMasc* much stronger than *Del1-OfMasc* (Supplementary Fig. 3f).

3. The AlphaFold2 analysis is okay, but it could also use validation. If the authors are confident of these structures, they could create mutants that should block binding or use directed crosslinking to validate their structures.

(Authors' reply)

We totally agree with this comment that AlphaFold2 analysis is just a prediction and only suggests the interaction of ankyrin repeats and Masc protein. Deletion analysis of ankyrin repeats of Oscar clearly revealed that the number of repeats are correlated with the binding (and inhibiting) activity against Masc (Supplementary Fig. 8b–d), supporting the AF2 prediction. Detailed structural analysis of this complex should be done by Cryo-EM analysis or other methods, which will identify the detailed conformation and essential residues for this interaction.

Minor:

1. Both Ref. 3 and 16 should be included for the discovery of the CifAB factors noted on line 60, p. 3.

(Authors' reply)

Corrected.

2. The authors several times state from proteasome inhibition that degradation by the ubiquitin-proteasome system is involved. However, no test of a ubiquitin requirement is done, and there are protein substrates that do not require it (although I doubt this is true here). This could just be written more carefully, or ubiquitin pathway inhibition could be tested using commercially available E1 ubiquitin activating-enzyme inhibitors (I think Takeda).

(Authors' reply)

We totally agree with this Reviewer's comment and described this point carefully in the revised manuscript. In addition, we examined the effect of MLN7243 (TAK-243), a selective ubiquitin activating enzyme UBA1 inhibitor, on OfMasc accumulation in *Wolbachia*-infected cells. As shown in the figure below (Fig. R1), we observed accumulation of OfMasc-GFP in OfT1C cells (*Wolbachia*-infected cells) treated with MLN7243 (1 μ M for 6 h or 300 nM for 24 h), suggesting that the ubiquitin-proteasome system is possibly involved in OfMasc degradation in the presence of *Wolbachia*. Because this result is preliminary, we decided not to include this figure (Fig. R1) in the manuscript. Further experiments will clarify the detailed mechanism of Masc degradation by *Wolbachia* infection.

Fig. R1. Effect of MLN7243 on OfMasc accumulation in *Wolbachia*-infected cells.

OfT1C cells were transfected with OfMasc-GFP. Two days after transfection, cells were treated with 1 μ M (a) or 300 nM (b) MLN7243, and GFP fluorescence was examined after 6 h and 24 h treatment, respectively.

- The authors note that a number of ubiquitin pathway enzymes have ankyrin repeats. Do they believe that Oscar is part of a ubiquitin ligase (E3) complex? This would make sense with respect to the effects on Masc levels. The deubiquitylase could have an editing function or could regulate the E3 activity. If the authors do an Oscar pulldown/MS analysis, do they identify other components of ubiquitin ligase complexes or pathways?

(Authors' reply)

We are also interested in this point. Our IP/LC-MS/MS experiments using GFP nanobody identified several ubiquitin pathway-related proteins, but at present it is unclear whether this binding is artifact or not. Considering that the CifB-like domain of Oscar is not essential for, but has a little effect on Oscar's function, we have to

proceed the research carefully whether Oscar is a component of this complex.

4. Although it is probably not critical to the current paper, I did not understand the evidence that Masc controls masculinization in the cytoplasm and dosage compensation in the nucleus. This part could be written more clearly.

(Authors' reply)

To make the manuscript clearly and easily, we deleted these sentences from the result section, which did not affect the story.

5. On line 241, the BmIMP(M) factor is mentioned but I'm not sure this was introduced earlier. Not sure where it fits in exactly.

(Authors' reply)

We added a brief description and a reference of *BmIMP* in the text.

We thank this Reviewer again for your helpful suggestions.

Reviewer #3 (Remarks to the Author):

Dear authors,

The manuscript by Katsuma et al. identifies and characterizes a novel Wolbachia factor involved in male-killing in *Ostrinia*. The authors established new cell-lines with and without Wolbachia and showed that these cell-lines are phenocopies of infected embryos. Using GFP-fused Masc they showed that accumulated of Masc is inhibited in Wolbachia-infected cells but not in uninfected cells. The interacting parts of Masc were tested with different induced deletions and the first zinc finger was shown to be involved in the binding to the Wolbachia factor. The Wolbachia factor was identified using a clever combination of the full-length GFP-Masc and the deleted GFP-Masc in a immunoprecipitation assay. The authors show that part of the novel Wolbachia protein, Oscar, contains a CifB terminus-like part which was shown to be involved in another form of reproductive manipulation in other insects. The authors present convincing data that indeed a male-killing factor has been functionally characterized for the first time. The evolution of this factor touched upon in the discussion but more phylogenetic analysis would be necessary to solidify this story. In our view the authors have shown an impressive amount of work that is of tremendous interest to the scientific community.

After revision (among which putting the work here in a more extended framework of previous research, improving the details in the materials and methods), and the addition of missing data (wFur genome) we think this manuscript is publishable.

Kind wishes,

Dr. Eveline Verhulst

Dr. Sander Visser

(Authors' reply)

We thank Prof. Verhulst and Dr. Visser for providing detailed comments.

Structural suggestions

Abstract

I would suggest the authors to remove the references in the abstract as this is unusual.

Please add titles "Introduction" and "Results" to the manuscript.

(Authors' reply)

This manuscript was directly transferred from another journal without any format changes.

We changed the manuscript style according to formatting instructions for this journal.

Specific comments to the text

Line 56-65: This section does not entirely do justice to the contents of Perlmutter et al. Especially the formulation in lines 64-67 suggest that wMel induces male-killing in *D. melanogaster*, which is not the case. Perlmutter and all indicate that they studied the wMel strain to find male-killing genes as they found that the 99,7% similar wRec strain can induce both male-killing and CI in different *Drosophila* species. The identified wmk gene has homologs in many male-killing *Wolbachia* strains as have been cifA and cifB, and are not just residing on wMel.

This line needs to be reformulated to indicate that wmk is a candidate for the male-killing phenotype of many *Wolbachia* strains.

(Authors' reply)

Corrected.

Also in the discussion the findings in this manuscript should be put much more into context of what was found in the study of Perlmutter et al. Is Oscar completely different from wmk? Does wFur also contain wmk if Oscar is not the same as wmk?

(Authors' reply)

wmk encodes a transcriptional regulator with two HTH domains (Perlmutter et al., 2019),

which is apparently different from Oscar (Fig. 2d). We described this clearly in the introduction and discussion sections of the revised manuscript.

We identified two *wmk* genes from the *wFur* genome, and investigated the functions of two *wmk* genes and the closest *wmk-like* gene in our cell-based assay. The results clearly showed that Wmk homologues do not inhibit Masc accumulation and Masc-induced masculinization in lepidopteran cells (Supplementary Fig. 11a–c).

We suggest the authors use this paper more in the discussion, and adjust the mentioned section in the introduction.

(Authors' reply)

We modified both introduction and discussion sections of the manuscript.

Line 69: suggestion to change “male-killing” to infected

Line 81: please replace “from” by “in”.

(Authors' reply)

Corrected.

Line 87: The reference to the figure in the context of this sentence is unclear to me. The importance and relevance of Fig 1a is also unclear in the context here. In addition the pictures are hard to interpret and not explained well in the caption. Are the black dots Wolbachia?

(Authors' reply)

We added arrows in this figure.

Lines 96-97 and line 102-104: The suggestion that Masc proteins have two roles (masculinization and dosage compensation) depending on their localization is not shown in this manuscript, but can also not be concluded (and has not been concluded) in the referenced article. Therefore, the writing here that the masculinizing function of Masc protein occurs in the cytoplasm is not valid and should be adjusted, or additional references should be added that can support this statement.

(Authors' reply)

To make the story clearly and easily, we deleted these sentences from the result section.

In general the logics of this paragraph (line 96-107) are hard to follow for researchers not intimately known with the details of *Ostrinia* previous work. It would be good to clarify that overexpression of OfMasc with cRNA was shown before to prevent male-killing but

that it was not enough to masculinize embryos. The description of it is now a bit cryptic.

(Authors' reply)

We deleted this sentence from the result section, which did not affect the story.

Line 99: please replace “survived” to “surviving”.

(Authors' reply)

As described above, this sentence was deleted.

Line 111 (Fig. 1b): What are the readers supposed to see in these two pictures? Are these pictures necessary for the main figure? We think they are better suited for the supplementary.

(Authors' reply)

We moved this figure to the Supplementary figures (Supplementary Fig. 1a).

Lines 114-115: default splicing is not shown in Figure 1e to which is referred here. Figure 1e shows splicing of *Ofdsx* after transfection with GFP and *OfMasc*-GFP.

(Authors' reply)

Added as Supplementary Fig. 1c (below panel).

Line 116: transfection with only *OfMasc* is not shown in Figure 1E, do the authors mean GFP here instead?

(Authors' reply)

Supplementary Fig. 1c shows the *Ofdsx* splicing transfected with *GFP* or *OfMasc* cDNA.

Please explain the cDNA here. What does it encode? Cannot find the description of these GFP-*OfMasc* fused protein in the material and methods either.

(Authors' reply)

We described a reference for *OfMasc*-*GFP* cDNA in the method section.

Line 118-119: Please explain why *OfMasc* expression did not affect the splicing pattern of *Ofdsx*. Is this again due to the lower levels of *OfMasc* not meeting the threshold necessary to induce the shift in splicing?

(Authors' reply)

This point should be explained after other experimental results are given. We described this in lines 124–128 of the revised manuscript.

Supplementary Fig. 1b is the same as Fig. 1e. Please check and adjust.

(Authors' reply)

We confirmed that they are different.

Lines 122-125: please change “origin is same” to “origin is the same”.

Line 148: please replace “for” by “in”.

(Authors' reply)

Corrected.

Lines 152-155: It is unclear from the description how the final candidate protein was identified from the 1045 initial proteins. Was there only a single protein derived from wFur recovered among the 1045 candidates and the additional 1044 proteins all originated from the host? The way this is currently written, it seems that there was only one wFur protein that interacted with the OfMasc N-terminal region suggesting there were more wFur proteins that were recovered but did not interact with the N-terminal which is confusing because it would require additional experiments to determine which region of the OfMasc protein is bound. Please clarify.

(Authors' reply)

In the revised version, we clearly described that the 1045 proteins are composed of both host and *Wolbachia* protein, and that among these proteins, only one protein Oscar was specifically and reproducibly detected in the immunoprecipitate bound to OfMasc-GFP, but markedly lower in that bound to Del1-OfMasc-GFP.

Line 194: It is difficult to see in the figure that OfMasc changes configuration. It can be seen that Oscar changes configuration, but OfMasc change in configuration is difficult. Is it possible to have a separated image of the configuration difference of OfMasc when captured by Oscar?

(Authors' reply)

We respectfully disagree. We think the center figure of Fig. 2e clearly shows disruption of alpha-helix of the Masc masculinizing domain. This was described in the text.

Line 196-199: This part of the paragraph has no links to the rest of the paragraph and its message is unclear. Please expand on this or move this section to a more appropriate location.

(Authors' reply)

We respectfully disagree. This sentence describes that *Oscar* sequence was not identified

in any *Wolbachia* genomes publicly available other than *wBol-1b*. We think the position of this sentence is appropriate.

Lines 204-206: Rather than a partial deletion in an ancestral *Wolbachia* of *wPip*, this could have just as well been an insertion/fusion in an ancestral lepidopteran *Wolbachia* lineage. A phylogeny of the different *Wolbachia* strains with the WO prophage would be helpful to put this in better context.

(Authors' reply)

wFur genome does not possess a typical WO prophage region and the high-quality genome sequences of male-killing *Wolbachia* for lepidopteran insects are not available at present. Thus the same supergroup *Wolbachia wPip* was used for this analysis.

Lines 230-232: I would recommend to rephrase this sentence a little as the expression level of *dsx* is likely not affected by Oscar, but the splicing of *dsx* is (indirectly) shifted from male- to female-specific. Therefore, stating that Oscar reduces the expression level of the male-type *dsx* variant is not fully accurate.

(Authors' reply)

Corrected.

Line 240: please change the second “expression” for “splicing”.

(Authors' reply)

Corrected.

Line 245: this statement is too strong. The *dsx* splicing data from the BmN-4 cells is showing small effects for some of the tested Masc proteins. More importantly, given the presented data, Oscar is able to reduce Masc accumulation, but no data is presented on the presence of Oscar in other *Wolbachia* species/strains in Lepidoptera, so stating that this is “strongly suggesting that Oscar commonly inhibits” this pathway is way too strong. Please tone down the statement to match the observed results.

(Authors' reply)

We toned down the statement.

Line 250: the use of expression here is confusing. We suggest to change these instances to splicing in the paragraph.

(Authors' reply)

Corrected.

Line 250-252: Were the hatched larvae injected with Oscar cRNA genetically or phenotypically female? Please add in the text.

(Authors' reply)

Fig. 4d shows genotyping results. We added "genetically" in the text.

In Fig. 4b: Why are the sample sizes for Oscar females and GFP males much lower than the other two groups?

(Authors' reply)

We randomly selected *B. mori* eggs used for injection experiments, thus their sexes are not equally divided.

In Fig. 4e: Why not indicate the male band as well? Indicating the female band is a bit odd here, as it's equally as important to also know what the other band is.

(Authors' reply)

This result shows the W chromosome-specific genotyping, which only identifies females. It is impossible to perform Z chromosome-specific gel-based genotyping, because both sexes possess Z chromosome (qPCR can also be used for male/female genotyping.).

In addition, are all the dead embryos shown in this picture? Meaning that only 6 died in the GFP group, but 16 died in the Oscar group? Please add this to the caption.

(Authors' reply)

Yes, the results showed all dead embryos in the experiment (Fig. 4a–d). *Oscar* cRNA injection resulted in production of more dead embryos, due to male killing. We added this to the legend.

Lines 269-271 and Fig. 4f: I would strongly recommend to remove the cytoplasm and nucleus sections, as this was not specifically shown in this study nor in the cited studies.

(Authors' reply)

Modified.

Lines 272-273: similar to the previous comment, this study shows that Masc is targeted by Oscar, but not that Oscar specifically affects dosage compensation. This needs to be elaborated upon and needs a reference, or, the statement needs to be toned down.

(Authors' reply)

Corrected.

Line 284: We don't agree that 2 cases of proteins with ankyrin repeats suggests that this is a common pathway through which sexual manipulation by symbiotic bacteria is established. This statement is worded too strongly, given the available data. Please tone it down.

(Authors' reply)

This is just a suggestion, and we think this is not a strong description. We slightly toned this down in the revised manuscript.

One point we are missing in the discussion is based on previous data published by the authors on the effect of wFur on the expression of OfMasc. In Fukui et al. 2015 (doi: 10.1371/journal.ppat.1005048), the authors found and confirmed that OfMasc expression is lower in Wolbachia-infected individuals than in non-infected individuals (see Figure 3 of that article). Given that Oscar interacts with OfMasc (and Masc of other lepidopteran species) on the protein level rather than the DNA or RNA level, we would like to ask the authors to discuss how the previous data fit with the current data in terms of expression levels.

(Authors' reply)

Added.

In addition, as mentioned above, please put the finding in this manuscript in the context of the wmk gene found in Perlmutter et al.

(Authors' reply)

Added.

Comments to Methods

In general

Please make sure to include the countries for all companies, and not only for few companies.

(Authors' reply)

Added.

Please make sure to remove the sequences from the material and methods, submit them to GenBank and add accession numbers in their place in the text.

(Authors' reply)

We deposited these sequences and added the GenBank numbers in the text.

Line 460: SFM stands for Serum Free Medium, so the use of the word “medium” behind SFM is redundant and should be removed. This is a repeated error, so please correct it throughout the manuscript.

(Authors' reply)

Corrected.

Lines 486-578: The details of the experiment are not described in this subsection to the extent that the experiments can be replicated. The starting concentration of RNA for cDNA conversion can make a difference in the final results and should be given. In addition, the conditions for the RT-PCR are not given, i.e. was a 2- or 3-step PCR cycle used (and if it was a 3-step, what annealing temperature was used?), the amount/volume of template cDNA is not given, the total reaction volumes are not given. The information given here is the same as in the four referred articles, therefore, this information can also not be found in those articles and should be included here.

(Authors' reply)

Added.

Line 495: Are these sequences that are predicted bioinformatically, or are these sequenced PCR fragments? Please describe that the fragments were sequenced (and the steps preceding this) if they were.

(Authors' reply)

Added.

Lines 496-568: These sequences don't belong in the Methods section and should rather be submitted to GenBank. Instead, the GenBank identifiers can be given here in the text.

(Authors' reply)

We deposited these sequences and added the GenBank numbers in the text.

Line 585: No primer sequences or reference given for the EF1 α primers.

(Authors' reply)

We added the corresponding reference.

Line 789: please correct to “a Wolbachia-infected.....”

(Authors' reply)

Corrected.

Line 792: details about the Illumina sequencing data are missing, i.e. specific Illumina platform, read size, insert size, single- or paired-end. Also, was there any software used for quality control of either the long or short read datasets before genome assembly and read mapping?

(Authors' reply)

Added.

In addition, where can the sequence of wFur data be found? This should be uploaded to GenBank and the identifier can be given here.

(Authors' reply)

We added the GenBank number.

Paragraph starting at line 819: Details regarding the PCR are missing, apart from the primers. In addition, there are several steps missing starting at line 832, i.e. RNA/DNA isolation, cDNA conversion, etc.

(Authors' reply)

Added.

Mentioned in line 833 that embryos were subjected to RT-PCR or molecular sexing, is this correct or were embryos used for both these experiments? In the figures of the manuscript it appears that the latter should be true. In that case, there is a description missing of DNA and RNA isolation of single embryos.

(Authors' reply)

Corrected.

Comments to figures

Fig. 1.

1a-b: I recommend to change the colour of “a” and “b” to black, or, if it improves visibility, add a white box around the black letter. The white font does not improve the visibility.

(Authors' reply)

We used black letters in figures.

Please also keep the placing of the scale bar consistent for all pictures, and add it in all pictures.

1d: please correct the scale bar, the original is also visible in the bottom left corner of

each picture.

1f: We recommend putting the same style thick scale bar on each picture because the narrow version is not properly visible (in addition the writing of the thin scale bar is still visible in the first picture, I would correct that).

Fig. 2.

2a: Similar issue with the scale bar as in 1f. In addition, the ends of the “original” scale bar are visible above the added scale bar in the first picture, revealing that the added scale bar is not exactly the same size as the original. This definitely should be fixed for all scale bars in all figures (it is for example also visible in figure 3a).

(Authors' reply)

We put scale bars in all pictures in Fig. 1e, Fig. 2a, Fig. 3a, Supplementary Fig. 1d, Supplementary Fig. 1f, Supplementary Fig. 3b, Supplementary Fig. 3d, Supplementary Fig. 7a, Supplementary Fig.7b, Supplementary Fig.8b, and Supplementary Fig.11b.

2e: The scales of the heatmap are not readable and axes have no description. Add a better description of these heat maps to the caption.

(Authors' reply)

Added.

Fig. 3.

3a: Different pictures not aligned properly. Please fix the scale bars here as well.

(Authors' reply)

Fixed.

Extended Data Fig. 8d: We are surprised to see that in none of the controls there is any variation between the 3 replicates, especially considering the data in Extended Data Fig. 8b. Therefore, I question how the controls were set to 100% (and how is the undoubtedly present variation between the 3 replicates incorporated)?. Is there any explanation why transfection with the different Masc-GFPs results in variation of BmIMPM splicing levels, but co-transfection with an empty vector, nullifies this variation. Was transfection done three times independently, or did the 3 samples originate from a single transfection experiment?

(Authors' reply)

As indicated in the y-axis and legends of these figures, the results are from three “independent” experiments (not replicates) and shown as the data compared with that of OfMasc-GFP (Supplementary Fig. 9b) or each Masc (Supplementary Fig. 9d).

Differences in *BmIMP^M* induction by each Masc protein have been reported in our previous studies (Lee et al., *Insect Mol. Biol.*, 2015; Fukui et al., *PLoS Pathogens*, 2015; Katsuma et al., *FEBS Open Bio*, 2019).

We thank the Reviewers again for their helpful suggestions.

Reviewer #2 (Remarks to the Author):

The authors have done a good job in addressing the reviewer questions and concerns, including my own. There are just a few relatively minor issues remaining:

1. The more thorough deletion analysis in Suppl. Fig. 8 that tested the various segments of Oscar for their effects on Masc levels and dsx splicing in cultured cells, including now the putative deubiquitylase domain, was a useful addition. However, the control Western of the expression levels of these deletion variants was missing, as far as I could tell. Therefore, it is possible that misfolding and poor expression, or overexpression, were responsible for some of the effects seen. This Western control needs to be added.

2. Also in Suppl. Fig. 8: It is puzzling that DelA no longer can suppress Masc expression but its effect on dsx-M splicing is statistically no different than full-length Oscar. Even DelB, which deletes almost half of the ANK repeats and also causes no drop in Masc levels anymore, still only has ~20% dsx-M relative to the vector control. These results would seem to argue against the main hypothesis of the paper regarding Oscar regulation of dsx splicing by controlling Masc levels. Could the authors please explain how they interpret these data?

3. Fig. 1A: Hard to tell what the arrows are pointing at. It almost looks like Wolbachia cells (?) are folded around host cytoplasm. What exactly is the evidence that these are bacterial cells? (I'm not sure this panel is even necessary.)

4. Lines 137-140: This part reads a little confusingly because the authors talk about Masc "interaction" before the Oscar interactor is identified. It would be more accurate to speak of differential Masc "accumulation" for the different alleles.

5. Line 179: I would change "forms" to "is predicted to form"

6. Line 200: "use them for male killing" to "is predicted to use them for male killing"

7. Line 224: I would change "indicated" to "suggested"

Reviewer #3 (Remarks to the Author):

I am happy with the revisions of the authors. The findings presented in the manuscript are of great scientific importance and conclusions are well supported by the results. All figures and explanations are clear and the methodology includes all relevant details. Concerning the data availability, the bioproject CP096925 is accessible, but all other accession numbers did not give a result when searching for them. Please check these.

Point-by-point responses to the reviewer's comments

Reviewer #2 (Remarks to the Author):

The authors have done a good job in addressing the reviewer questions and concerns, including my own. There are just a few relatively minor issues remaining:

(Authors' reply)

We thank this Reviewer for his/her encouraging comments.

1. The more thorough deletion analysis in Suppl. Fig. 8 that tested the various segments of Oscar for their effects on Masc levels and dsx splicing in cultured cells, including now the putative deubiquitylase domain, was a useful addition. However, the control Western of the expression levels of these deletion variants was missing, as far as I could tell. Therefore, it is possible that misfolding and poor expression, or overexpression, were responsible for some of the effects seen. This Western control needs to be added.

(Authors' reply)

As pointed out by this Reviewer, we examined the expression levels of Oscar derivatives by Western blotting using an in-house Oscar antibody. We observed similar accumulation levels of DelA and DelB derivatives compared to that of wild-type Oscar (Supplementary Fig. 8b). However, we did not detect the expression of C-terminus-lacking DelC and DelD derivatives with anti-Oscar antibody, because this antibody was raised against a synthetic peptide corresponding to the C-terminus of Oscar. To investigate the expression levels of these two derivatives, we generated additional 3×FLAG-tagged Oscar constructs (FLAG-DelC and FLAG-DelD) and successfully detected their expressions using anti-FLAG antibody (Supplementary Fig. 8c). In addition, we obtained similar results of *Sf*dsx^M expression in Sf9 cells transfected with FLAG-tagged Oscar constructs (Supplementary Fig. 8f) when observed in FLAG-less Oscar transfection (Supplementary Fig. 8e). These results support our conclusion that ankyrin repeats and linker region are crucial for Oscar's functions.

2. Also in Suppl. Fig. 8: It is puzzling that DelA no longer can suppress Masc expression but its effect on dsx-M splicing is statistically no different than full-length Oscar. Even DelB, which deletes almost half of the ANK repeats and also causes no drop in Masc levels anymore, still only has ~20% dsx-M relative to the vector control. These results would seem to argue against the main hypothesis of the paper regarding Oscar regulation of dsx splicing by controlling Masc levels. Could the authors please explain how they interpret these data?

(Authors' reply)

Thank you for providing critical comments. We think that both DelA and DelB bind Masc and inhibit its functions, but these derivatives lose the ability to degrade the Masc protein. These points are important to understand the mechanism of Oscar-mediated male killing. We described this in the revised manuscript.

3.Fig. 1A: Hard to tell what the arrows are pointing at. It almost looks like Wolbachia cells (?) are folded around host cytoplasm. What exactly is the evidence that these are bacterial cells? (I'm not sure this panel is even necessary.)

(Authors' reply)

We removed this TEM figure from the manuscript.

4.Lines 137-140: This part reads a little confusingly because the authors talk about Masc "interaction" before the Oscar interactor is identified. It would be more accurate to speak of differential Masc "accumulation" for the different alleles.

(Authors' reply)

Thank you for your suggestion. We modified this part in the revised manuscript.

5.Line 179: I would change "forms" to "is predicted to form"

6.Line 200: "use them for male killing" to "is predicted to use them for male killing"

7.Line 224: I would change "indicated" to "suggested"

(Authors' reply)

Corrected.

Thank you again for your helpful suggestions.

Reviewer #3 (Remarks to the Author):

I am happy with the revisions of the authors. The findings presented in the manuscript are of great scientific importance and conclusions are well supported by the results. All figures and explanations are clear and the methodology includes all relevant details. Concerning the data availability, the bioproject CP096925 is accessible, but all other accession numbers did not give a result when searching for them. Please check these.

(Authors' reply)

We thank Prof. Verhulst and Dr. Visser for reviewing the manuscript very carefully. We confirmed that all deposited data and sequences are accessible now.

Reviewer #2 (Remarks to the Author):

I am fully satisfied with the response to my last, relatively minor comments and look forward to seeing this excellent paper published.

Point-by-point responses to the reviewer's comments

Reviewer #2 (Remarks to the Author):

I am fully satisfied with the response to my last, relatively minor comments and look forward to seeing this excellent paper published.

(Authors' reply)

We thank this Reviewer for reviewing the manuscript very carefully and his/her encouraging comments.